# Visual Autoregressive Modeling for Instruction-Guided Image Editing

**Qingyang Mao**[1][*], **Qi Cai**[2], **Yehao Li**[2], **Yingwei Pan**[2][†], **Mingyue Cheng**[1], **Ting Yao**[2], **Qi Liu**[1][†], **Tao Mei**[2]
[1]University of Science and Technology of China, [2]HiDream.ai Inc.
maoqy0503@mail.ustc.edu.cn, {mycheng, qiliuql}@ustc.edu.cn,
{cqcaiqi, liyehao, pandy, tiyao, tmei}@hidream.ai

## Abstract

Recent advances in diffusion models have brought remarkable visual fidelity to instruction-guided image editing. However, their global denoising process inherently entangles the edited region with the entire image context, leading to unintended spurious modifications and compromised adherence to editing instructions. In contrast, autoregressive models offer a distinct paradigm by formulating image synthesis as a sequential process over discrete visual tokens. Their causal and compositional mechanism naturally circumvents the adherence challenges of diffusion-based methods. In this paper, we present VAREdit, a visual autoregressive (VAR) framework that reframes image editing as a next-scale prediction problem. Conditioned on source image features and text instructions, VAREdit generates multi-scale target features to achieve precise edits. A core challenge in this paradigm is how to effectively condition the source image tokens. We observe that finest-scale source features cannot effectively guide the prediction of coarser target features. To bridge this gap, we introduce a Scale-Aligned Reference (SAR) module, which injects scale-matched conditioning information into the first self-attention layer. VAREdit demonstrates significant advancements in both editing adherence and efficiency. On EMU-Edit and PIE-Bench benchmarks, VAREdit outperforms leading diffusion-based methods by a substantial margin in terms of both CLIP and GPT scores. Moreover, VAREdit completes a 512×512 editing in 1.2 seconds, making it 2.2× faster than the similarly sized UltraEdit. Code is available at: `https://github.com/HiDream-ai/VAREdit`.

## 1 Introduction

Instruction-guided image editing (Brooks et al., 2023; Zhang et al., 2023; Sheynin et al., 2024) represents a significant advance in generative AI, shifting the paradigm from pure synthesis towards fine-grained interactive control. This progress has been propelled by two primary factors: the curation of large-scale, high-quality editing datasets (Ge et al., 2024; Ye et al., 2025) and the rise of diffusion models (Ho et al., 2020; Peebles et al., 2023; Labs, 2024) as the mainstream architecture. In particular, the impressive visual fidelity is largely derived from the iterative denoising process in diffusion models. However, this core mechanism also imposes a critical limitation. The global nature of the denoising process often leads to unintended spurious edits, causing modifications to "bleed" into regions that should remain unchanged. Simultaneously, the multi-step denoising process incurs substantial computational cost and limits real-time practical use.

In contrast to diffusion models, autoregressive (AR) models (Esser et al., 2021; Sun et al., 2024; Pang et al., 2025) offer a fundamentally different paradigm. Instead of holistic iterative refinement, they synthesize an image causally in a sequential token-by-token manner. This compositional generation process is a natural fit for image editing. Specifically, it provides a flexible mechanism for preserving unchanged regions while precisely modifying edited regions, addressing the entanglement problem of diffusion models. More encouragingly, the pioneering visual autoregressive (VAR)

---

[*]This work was performed at HiDream.ai.
[†]Corresponding authors.

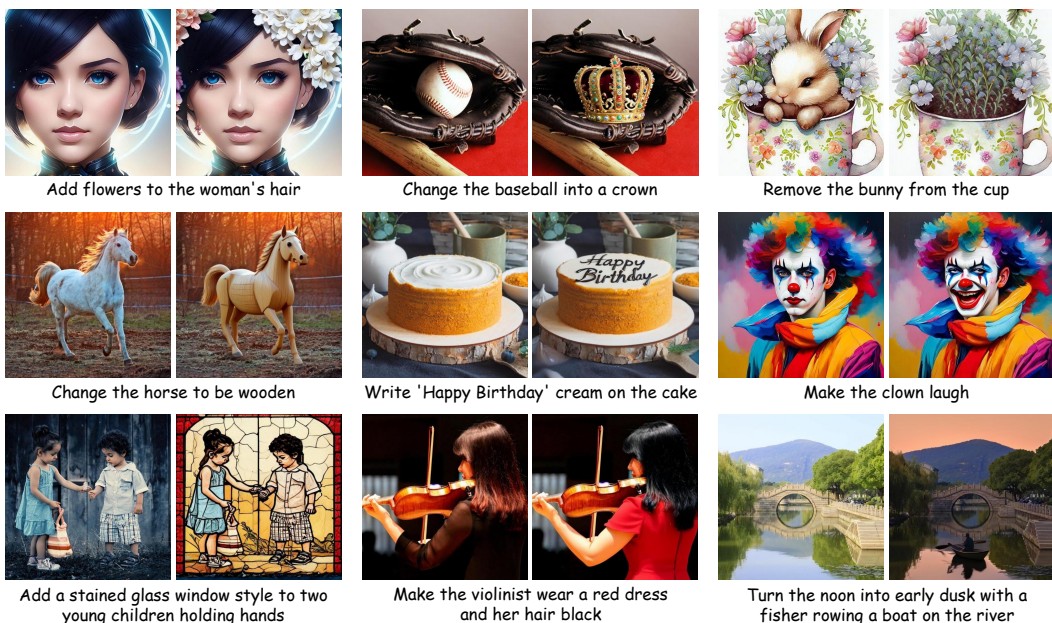

Figure 1: **VAREdit** achieves high-precision performance in instruction-guided image editing. It excels across diverse editing scenarios, including object-level modifications (addition, replacement, removal), attribute changes (material, text, posture, style, color) and complex compositional edits.

modeling (Tian et al., 2024) breaks the spatial structural degradation through the scale-by-scale prediction strategy, leading to high-quality image generation. While the VAR paradigm has shown competitive performance in image synthesis (Kumbong et al., 2025; Han et al., 2025), its potential for instruction-guided image editing remains largely unexplored. Early training-free attempts (Wang et al., 2025) repurpose text-to-image models but lack the task-specific knowledge required for robust editing, causing them to lag significantly behind diffusion-based methods (Hou et al., 2024; Zhao et al., 2024a; Zhang et al., 2025b).

We address this critical gap by introducing VAREdit, a novel tuning-based VAR framework for instruction-guided image editing. VAREdit reframes image editing as a next-scale prediction problem, which generates multi-scale target features to achieve precise image modification. A central challenge in this paradigm is how to effectively condition the model on the source image. A straightforward approach is to use full-scale source conditions, yet it is computationally inefficient due to the lengthy input sequences. While the finest-scale-only conditional setting provides sufficient information and eliminates the redundant input tokens, it creates a severe scale mismatch since high-frequency details may disrupt the prediction of coarse target features. To resolve this issue, we perform a systematic analysis of inter-scale dependencies and uncover a key insight: the model's sensitivity to scale-aware conditioning is concentrated in its first self-attention layer. Accordingly, we propose the Scale-Aligned Reference (SAR) module, a mechanism that injects the scale-matched source information only in the first self-attention layer, while other layers still use the finest-scale condition. As demonstrated in Figure 1, this design enables VAREdit to achieve exceptionally precise editing across diverse challenging editing scenarios. Our contributions are three-fold:

- We introduce VAREdit, a novel tuning-based visual autoregressive model for instruction-guided image editing. To the best of our knowledge, we are among the first to introduce next-scale visual prediction into instruction-guided image editing.

- We identify a scale-mismatch issue in the finest-scale conditioned VAREdit and propose the SAR module as an effective solution.

- VAREdit attains a new record on standard editing benchmarks, outperforming leading diffusion models in both editing adherence and generation efficiency.

## 2 RELATED WORK

### 2.1 INSTRUCTION-GUIDED IMAGE EDITING

Instruction-guided image editing (Brooks et al., 2023; Zhao et al., 2024a; Yu et al., 2025; Cai et al., 2025) seeks to modify an image according to user's textual instruction, which differs from caption-based approaches (Hertz et al., 2022; Feng et al., 2025) that often resynthesize an entire image from a global description. This field is predominantly driven by diffusion models and seminal work InstructPix2Pix (Brooks et al., 2023) established a powerful paradigm by channel-wise concatenating source and target images. Subsequent research advanced this by curating larger-scale datasets with "expert" models (Zhang et al., 2023; Sheynin et al., 2024; Zhao et al., 2024a; Ge et al., 2024; Ye et al., 2025) or exploring alternative conditioning, like spatial concatenation for large-motion edits (Zhang et al., 2025b; Cai et al., 2025). However, diffusion-based models often produce unintended modifications and suffer from slow iterative sampling. To address these limitations, recent studies have widely explored autoregressive (AR) models for instruction-guided image editing. Some approaches leverage the capabilities of Multimodal Large Language Models (MLLMs) in high-quality semantical alignment (Fu et al., 2024; Zhou et al., 2025) for further diffusion-based visual generation processes. Other methods employ AR-based frameworks directly for the image editing task, integrating knowledge distillation (Mu et al., 2025), mask-aware region conditioning (Wu et al., 2025b) or in-context reasoning for interactive manipulation (Lai et al., 2025). While these methods have achieved promising results, they still follow the vanilla next-token prediction paradigm with potential risks of structural degradation. Our work builds on the pioneering visual autoregressive (VAR) (Tian et al., 2024) architecture that follows the next-scale prediction paradigm, a tuning-based approach that surpasses recent approaches in both edit quality and efficiency.

### 2.2 AUTOREGRESSIVE IMAGE GENERATION

Autoregressive (AR) models, a mainstay in language tasks, have been extended to image synthesis. They operate by tokenizing an image into a discrete sequence and then training a Transformer model to predict the next token. Early works use vector quantization (VQ) for tokenization (Van Den Oord et al., 2017; Esser et al., 2021; Chang et al., 2022; Yu et al., 2022), while recent advances such as lookup-free quantization (Yu et al., 2023; Zhao et al., 2024b) have improved performance with large vocabularies. The AR modeling paradigm itself has also made significant progress. For instance, MAR (Li et al., 2024) uses masked multi-token prediction to improve quality and speed, while RandAR (Pang et al., 2025) enables flexible token ordering for new capabilities such as resolution extrapolation. To better capture spatial dependencies, models like VAR (Tian et al., 2024) and Infinity (Han et al., 2025) introduced coarse-to-fine prediction schemes. As AR approaches are well-suited for language modeling, they demonstrate a significant advantage in instruction-following performance (Lai et al., 2025; Zhang et al., 2025a) for image generation. Meanwhile, recent advances like hybrid parallelization and KV-scale mitigate the efficiency bottleneck (Kumbong et al., 2025; Li et al., 2025). These innovations are enhancing both generation quality and efficiency, enabling AR models to emerge as strong competitors to diffusion models, particularly in tasks requiring high fidelity to complex, compositional prompts.

## 3 METHODOLOGY

We first review the visual autoregressive (VAR) modeling paradigm. Then, we introduce VAREdit, a novel framework that reframes instruction-guided image editing as a multi-scale conditional generation task. Finally, we analyze the challenges of various source image conditioning strategies and present the inspiration and the design of our Scale-Aligned Reference (SAR) module, a targeted solution to the scale-mismatch problem in naive conditioning.

### 3.1 PRELIMINARY

Visual autoregressive (VAR) model generally comprises a multi-scale visual tokenizer and a Transformer-based generation model. The process begins with an encoder $\mathcal{E}$ that maps an image $\mathbf{I}$ to a continuous feature representation $\mathbf{F} = \mathcal{E}(\mathbf{I}) \in \mathbb{R}^{h \times w \times d}$. A quantizer $\mathcal{Q}$ then decomposes $\mathbf{F}$ into a hierarchy of $K$ discrete residual maps $\mathbf{R}_{1:K} = \mathcal{Q}(\mathbf{F})$. These maps follow a coarse-to-fine

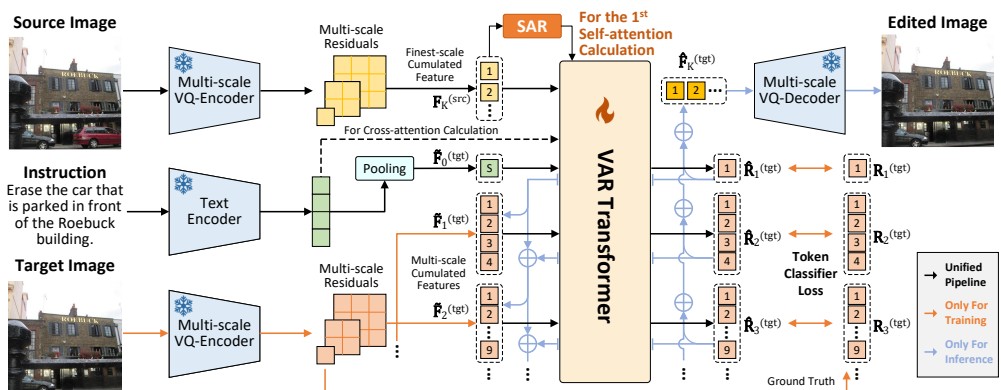

Figure 2: The overall architecture of VAREdit for instruction-guided image editing. VAREdit first encodes the images into multi-scale residuals by a shared vector quantizer and maps the instructions into textual token embeddings. These features are organized as the finest-scale source feature $\mathbf{F}_K^{(src)}$, the pooled textual representation $\widetilde{\mathbf{F}}_0^{(tgt)}$, the coarse-to-fine target features $\widetilde{\mathbf{F}}_{1:K-1}^{(tgt)}$, and then sent to the VAR Transformer. The source feature $\mathbf{F}_K^{(src)}$ is further sent to the Scale-Aligned Reference (SAR) module in the first self-attention layer, while the textual token embeddings are also used for cross-attention calculations of key and value matrices. The ground truth residuals $\mathbf{R}_{1:K}^{(tgt)}$ guide the training of the last $K$ output residuals $\hat{\mathbf{R}}_{1:K}^{(tgt)}$. During inference, the residuals $\hat{\mathbf{R}}_{1:K}^{(tgt)}$ are predicted autoregressively, which are then aggregated to $\hat{\mathbf{F}}_K^{(tgt)}$ and then decoded to the edited image.

structure, where the spatial resolution $(h_k, w_k)$ of each residual map $\mathbf{R}_k$ increases with its scale index $k$. Then, the Transformer model predicts the residuals in an autoregressive manner (where the scale features are flattened to obtain the sequential representation):

$$p(\mathbf{R}_{1:K}) = \prod_{k=1}^{K} p(\mathbf{R}_k | \mathbf{R}_{1:k-1}). \tag{1}$$

Concretely, to predict the residual map for the next scale $\mathbf{R}_{k+1}$, the model first computes an intermediate feature representation $\mathbf{F}_k$ by aggregating all previously generated residuals $\mathbf{R}_{1:k}$:

$$\mathbf{F}_k = \sum_{i=1}^{k} \mathrm{Up}(\mathrm{Lookup}(\mathbf{R}_i, C), (h, w)), \tag{2}$$

where $\mathrm{Lookup}(\cdot, C)$ represents retrieving vector embeddings from the learned codebook $C$ and $\mathrm{Up}(\cdot, \cdot)$ denotes upsampling operation (*e.g.*, bilinear upsampling). This cumulative feature $\mathbf{F}_k$ is then downsampled (*e.g.*, average pooling) to match the spatial dimensions of the next scale, $(h_{k+1}, w_{k+1})$, creating the input for the next scale's prediction step: $\widetilde{\mathbf{F}}_k = \mathrm{Down}(\mathbf{F}_k, (h_{k+1}, w_{k+1}))$. The process is initiated with $\widetilde{\mathbf{F}}_0$, a start-of-sequence representation derived from a conditional embedding (e.g., class labels). Once all $K$ residual maps $\hat{\mathbf{R}}_{1:K}$ have been generated, they are used to compute the final feature map $\hat{\mathbf{F}} = \hat{\mathbf{F}}_K$ through Equation 2. Finally, a decoder $\mathcal{D}$ synthesizes the output image $\hat{\mathbf{I}} = \mathcal{D}(\hat{\mathbf{F}})$ as the predicted generation.

## 3.2 VAREDIT

We introduce VAREdit, a framework that reframes instruction-guided image editing as a conditional, multi-scale prediction problem. Figure 2 presents the overall architecture of VAREdit, which leverages a pre-trained VAR model as its foundation and autoregressively generates the target residual maps $\mathbf{R}^{(tgt)}$ conditioned on a source image $\mathbf{I}^{(src)}$ and a textual instruction $\mathbf{t}$:

$$p(\mathbf{R}_{1:K}^{(tgt)} | \mathbf{I}^{(src)}, \mathbf{t}) = \prod_{k=1}^{K} p(\mathbf{R}_k^{(tgt)} | \mathbf{R}_{1:k-1}^{(tgt)}, \mathbf{I}^{(src)}, \mathbf{t}) = \prod_{k=1}^{K} p(\mathbf{R}_k^{(tgt)} | \mathbf{F}_{1:k-1}^{(tgt)}, \mathbf{I}^{(src)}, \mathbf{t}), \tag{3}$$

where the second equation comes from the fact that $\mathbf{F}_k^{(tgt)}$ integrates all previous $k$ scales' residual information $\mathbf{R}_{1:k}^{(tgt)}$ according to Equation 2 and we take the spatial-dimension aligned form $\widetilde{\mathbf{F}}_k^{(tgt)}$ of the aggregated features as input. All scale features are flattened to obtain a sequential representation.

### 3.2.1 CONDITIONING SOURCE IMAGES

As for the specific design of VAREdit, the key challenge is how to effectively and efficiently incorporate the source image $\mathbf{I}^{(src)}$ to guide the multi-scale generation process.

**Full-scale Conditioning**. A straightforward approach is to condition the generation on the complete multi-scale features of the source image, $\mathbf{F}_{1:K}^{(src)}$. This is achieved by prepending the sequence of all source tokens to the target sequence, which allows the model to reference any scale source residual when generating the target residual. The conditional likelihood becomes:

$$p(\mathbf{R}_{1:K}^{(tgt)}|\mathbf{I}^{(src)}, \mathbf{t}) = p(\mathbf{R}_{1:K}^{(tgt)}|\mathbf{F}_{1:K}^{(src)}, \mathbf{t}) = \prod_{k=1}^{K} p(\mathbf{R}_k^{(tgt)}|\mathbf{F}_{1:k-1}^{(tgt)}, \mathbf{F}_{1:K}^{(src)}, \mathbf{t}). \tag{4}$$

While this method provides a comprehensive, scale-by-scale reference for the editing task, it is computationally expensive. Doubling the sequence length results in a quadratic increase $O(n^2)$ in the cost of self-attention, rendering it impractical for high-resolution editing. Moreover, providing multiple source scale features may introduce redundant or conflicting information for target feature prediction, potentially degrading editing quality.

**Finest-scale Conditioning**. To address the prohibitive cost of full-scale conditioning, we propose a more efficient strategy to condition solely on the finest-scale source feature, $\mathbf{F}_K^{(src)}$. This approach is motivated by the hierarchical nature of the visual tokenizer: the finest scale encapsulates the most detailed, high-frequency information from the source image, which is often the most critical for guiding an edit. This simplification slims down the likelihood to:

$$p(\mathbf{R}_{1:K}^{(tgt)}|\mathbf{I}^{(src)}, \mathbf{t}) = p(\mathbf{R}_{1:K}^{(tgt)}|\mathbf{F}_K^{(src)}, \mathbf{t}) = \prod_{k=1}^{K} p(\mathbf{R}_k^{(tgt)}|\mathbf{F}_{1:k-1}^{(tgt)}, \mathbf{F}_K^{(src)}, \mathbf{t}). \tag{5}$$

In this way, only the tokens from $\mathbf{F}_K^{(src)}$ are prepended to the target sequence. While this strategy dramatically reduces sequence length and thus alleviates the computational bottleneck of the vanilla full-scale conditional setting, it introduces a critical *scale mismatch* challenge. The model is tasked with predicting coarse-grained target image structure while having access to only the fine-grained, local details in the source condition, which is insufficient for precise editing.

### 3.2.2 SCALE DEPENDENCY ANALYSIS

The critical scale mismatch challenge introduced by the efficient finest-scale approach raises a fundamental question: *which source scales are truly necessary for high-fidelity editing?* To investigate such scale dependencies of target residuals and source residuals, we perform a diagnostic analysis of the self-attention mechanisms in a model trained on the *full-scale* source features. This full-scale setting allows the model to freely attend to all source scales.

The self-attention heatmaps in Figure 3, averaged across all heads and samples from PIE-Bench (Ju et al., 2023), reveal that different Transformer layers develop distinct attention patterns. In the first self-attention layer, when predicting tokens for a given target scale, the attention mechanism distributes broadly, focusing heavily on the corresponding and all *coarser* source scales. This pattern indicates that the initial layer is responsible for establishing the global layout and long-range dependencies. This behavior shifts in deeper layers, where attention patterns become highly localized. They exhibit strong diagonal structures, suggesting that attention is primarily confined to tokens in the spatial neighborhood. This functional transition suggests a shift from global structuring to local refinement, where the model copies and adjusts high-frequency details. For this latter task, the fine-grained information provided by $\mathbf{F}_K^{(src)}$ is sufficient. This motivates us to design a hybrid solution that provides the scale-aligned reference in the first layer while all subsequent layers attend only to the finest-scale source. See Appendix A.7 for self-attention heatmaps of all model layers.

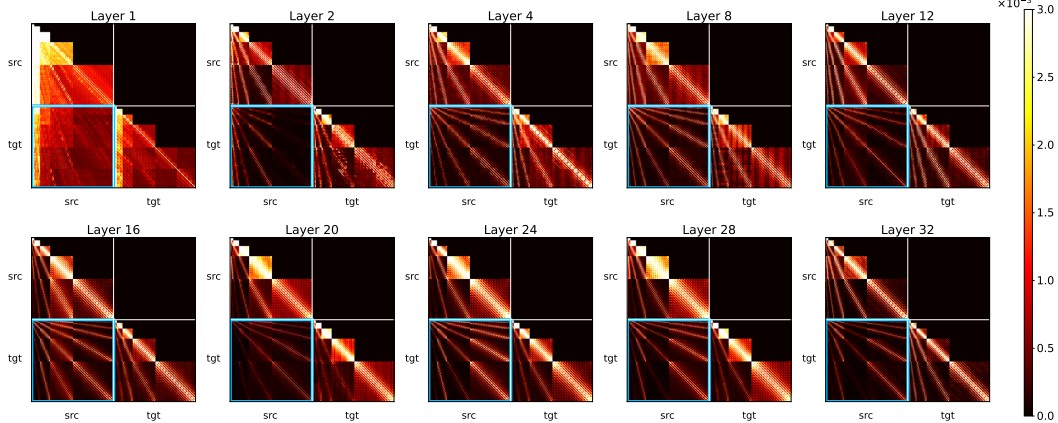

Figure 3: Self-attention heatmaps of PIE-Bench samples based on the full-scale transformer across different layers. The dependency patterns differ in the first self-attention layer versus others, inspiring a scale-aligned reference module.

### 3.2.3 SCALE-ALIGNED REFERENCE

Based on our analysis, we introduce the Scale-Aligned Reference (**SAR**) module, designed specifically to resolve the scale mismatch problem within the first self-attention layer. The core idea is to dynamically generate coarse-scale reference features by downsampling the single, finest-scale source feature map $\mathbf{F}_K^{(src)}$. This creates a set of reference features, each aligned to the spatial dimensions of a target scale:

$$\mathbf{F}_k^{(ref)} = \text{Down}(\mathbf{F}_K^{(src)}, (h_k, w_k)). \tag{6}$$

During generation, when predicting the tokens for target scale $k$, the first self-attention layer computes queries $\mathbf{Q}_k^{(tgt)}$ and attends to a combined set of keys and values, which are derived from two sources: (i) the newly generated, scale-aligned reference feature $\mathbf{F}_k^{(ref)}$ and (ii) the causal history of previously generated target tokens. Specifically, the self-attention output $\hat{\mathbf{O}}_k^{(tgt)}$ is computed as:

$$\hat{\mathbf{O}}_k^{(tgt)} = \text{Softmax}\left(\frac{\mathbf{Q}_k^{(tgt)}\left[\mathbf{K}_k^{(ref)\top}, \mathbf{K}_1^{(tgt)\top}, \cdots, \mathbf{K}_k^{(tgt)\top}\right]}{\sqrt{d}}\right) \cdot \left[\mathbf{V}_k^{(ref)\top}, \mathbf{V}_1^{(tgt)\top}, \cdots, \mathbf{V}_k^{(tgt)\top}\right]^{\top},$$

$$\tag{7}$$

where $(\mathbf{K}_k^{(ref)}, \mathbf{V}_k^{(ref)})$ are projected from the scale-aligned source reference $\mathbf{F}_k^{(ref)}$, and $(\mathbf{K}_{1:k-1}^{(tgt)}, \mathbf{V}_{1:k-1}^{(tgt)})$ are from previous target scales. This method reduces the spatial size of attention scores before the softmax operation without changing the output sequence length. Crucially, this SAR mechanism is applied *only to the first self-attention layer*. All subsequent layers operate using only the finest-scale conditioning, attending to $\mathbf{F}_K^{(src)}$ and the target's history residuals. This allows VAREdit to capture crucial scale dependencies in an aligned manner while achieving the efficiency of the finest-scale approach.

## 4 EXPERIMENTS

### 4.1 EXPERIMENTAL SETUP

**Datasets**. VAREdit is trained on a large-scale dataset of 3.92 million paired examples, aggregated from the SEED-Data-Edit (Ge et al., 2024) and ImgEdit (Ye et al., 2025) datasets. From the SEED-Data-Edit dataset, we extracted all single-turn samples and decomposed multi-turn conversations into single-turn editing pairs. These pairs were then filtered using a vision-language model (Team et al., 2025) to remove instances with poor instruction-following quality. Finally, all single-turn samples from ImgEdit were incorporated. Additional details regarding this data processing pipeline

are available in Appendix A.1. For evaluation, we assess VAREdit on four established benchmarks: (1) EMU-Edit (Sheynin et al., 2024), including 3,589 samples across eight distinct editing types, (2) PIE-Bench (Ju et al., 2023), including 700 samples that cover ten different editing types, (3) ImgEdit-Bench (Ye et al., 2025), including 737 samples in the basic-edit suite covering nine editing types and 47 complex samples in the understanding-grounding-editing suite, and (4) GEdit-Bench (English subset) (Liu et al., 2025), including 606 samples covering eleven extensive editing types.

**Evaluation Metrics**. Standard benchmarks like EMU-Edit and PIE-Bench rely on CLIP-based scores (Radford et al., 2021). EMU-Edit employs caption-image similarity (CLIP-Out.) and text-image directional similarity (CLIP-Dir.), whereas PIE-Bench measures similarity for the whole image (CLIP-Whole) and the edited region (CLIP-Edit). However, these metrics often fail to capture important editing quality aspects, such as spurious modifications or incomplete edits. To address these shortcomings, we also adopt the evaluation protocol from OmniEdit (Wei et al., 2025), which uses GPT-4o (Hurst et al., 2024) as an automated judge to provide two key scores on a scale of 0-10: (1) **GPT-Success** (Suc.): Measures the adherence to the editing instruction (higher is better) and (2) **GPT-Overedit** (Over.): Assesses the preservation of unedited regions (higher is better). Since a model could achieve a perfect GPT-Over. score by simply disregarding the edit instruction and outputting the original image, we further introduce **GPT-Balance** (Bal.) as an evaluation metric for overall editing performance, defined as the harmonic mean of the GPT-Suc. and GPT-Over. scores. Detailed prompts and computation methods, along with evaluation details about ImgEdit-Bench and GEdit-Bench, are provided in Appendix A.2.

**Compared Approaches**. To ensure a comprehensive and rigorous evaluation, we assess VAREdit against several state-of-the-art tuning-based methods. Our comparative analysis includes a broad spectrum of basic open-source methods: InstructPix2Pix (Brooks et al., 2023), UltraEdit (Zhao et al., 2024a), OmniGen (Xiao et al., 2025), AnySD (Yu et al., 2025), EditAR (Mu et al., 2025), ACE++ (Mao et al., 2025), ICEdit (Zhang et al., 2025b). We also include four frontier image editing approaches for comparison: GPT-4o-Image (Hurst et al., 2024), Step1X-Edit (Liu et al., 2025), FLUX.1 Kontext (dev) (Labs et al., 2025) and Qwen-Image-Edit (Wu et al., 2025a). The detailed descriptions of these compared approaches are provided in Appendix A.3.

**Implementation Details**. Our VAREdit models are initialized with weights from the pre-trained Infinity model (Han et al., 2025) along with its bitwise multi-scale residual quantizer (Zhao et al., 2024b). To differentiate source image tokens from target image tokens, we introduce a positional offset of $\Delta = (64, 64)$ to the 2D Rotary Position Embeddings (2D-RoPE) for all source tokens. To investigate scaling properties, we develop two distinct model sizes: VAREdit-2B and VAREdit-8B. VAREdit-2B undergoes a two-stage training procedure: an initial 8k iterations trained at $256 \times 256$ resolution with batch size 1,536 and learning rate $6e - 5$, followed by 7k fine-tuning iterations at $512 \times 512$ with batch size 960 and learning rate $1.875e - 5$. The larger VAREdit-8B is trained directly at $512 \times 512$ resolution for 60k iterations with batch size 1,536 and learning rate $6e - 5$. During training, we optimize the bitwise classifier loss over target residual token indices at each scale, following Infinity (Han et al., 2025). As for inference, we employ a classifier-free guidance (CFG) strength of $\eta = 4$ and a logits temperature of $\tau = 0.5$. Please refer to Appendix A.4 for more implementation details.

## 4.2 QUANTITATIVE RESULTS

**Superior Editing Quality**. The quantitative results in Table 1 demonstrate the superiority of VAREdit in editing performance. Among basic open-source methods, VAREdit consistently outperforms all diffusion-based and autoregressive baselines on our primary metric, GPT-Balance. Our VAREdit-8B model achieves a GPT-Bal. score of 7.892 on EMU-Edit and 8.105 on PIE-Bench, surpassing the strongest competitor (ICEdit on EMU-Edit, UltraEdit on PIE-Bench) by 64.9% and 45.3%, respectively. This underscores VAREdit's ability to perform precise edits while preserving unchanged regions. On conventional CLIP-based metrics, VAREdit also consistently outperforms these baselines and secures the top scores on both EMU-Edit and PIE-Bench. Compared to the four frontier approaches, GPT-4o-Image performs best on both the two benchmarks due to the large scale of training data and model size. Among the open-source approaches, VAREdit-8B performs better than Step1X-Edit and FLUX.1 Kontext (dev) according to GPT-scores and a slightly lower than Qwen-Image-Edit (20B). We have further evaluated the performance of VAREdit on GEdit-Bench. Please refer to Appendix A.6 to access the detailed results and analysis.

Table 1: Quantitative results of VAREdit, leading open-source methods and frontier image editing approaches on EMU-Edit and PIE-Bench benchmarks.

| Method | Size | EMU-Edit CLIP Score Out. | Dir. | GPT Score Suc. | Over. | Bal. | PIE-Bench CLIP Score Whole | Edit | GPT Score Suc. | Over. | Bal. | Time |
|---|---|---|---|---|---|---|---|---|---|---|---|---|
| InstructPix2Pix | 1.1B | 0.268 | 0.083 | 3.358 | 6.299 | 2.923 | 0.236 | 0.217 | 4.794 | 6.534 | 4.034 | 3.5s |
| UltraEdit | 7.7B | 0.278 | 0.095 | 4.881 | 7.704 | 4.541 | 0.247 | 0.220 | 5.831 | 8.350 | 5.580 | 2.6s |
| OmniGen | 3.8B | 0.274 | 0.082 | 4.738 | **8.709** | 4.666 | 0.240 | 0.212 | 3.459 | **8.939** | 3.498 | 16.5s |
| AnySD | 2.9B | 0.268 | 0.059 | 3.098 | 8.590 | 3.129 | 0.227 | 0.206 | 3.456 | 7.806 | 3.326 | 3.4s |
| EditAR | 0.8B | 0.273 | 0.064 | 3.582 | 7.260 | 3.305 | 0.242 | 0.216 | 5.070 | 8.116 | 4.707 | 45.5s |
| ACE++ | 17B | 0.258 | 0.058 | 2.375 | 5.979 | 2.076 | 0.234 | 0.205 | 2.743 | 8.093 | 2.574 | 5.7s |
| ICEdit | 17B | 0.271 | 0.095 | 5.027 | 7.591 | 4.785 | 0.234 | 0.215 | 5.321 | 7.593 | 4.933 | 8.4s |
| **VAREdit** (256px) | 2.2B | 0.268 | 0.091 | 6.072 | 7.058 | 5.565 | 0.256 | 0.224 | 7.234 | 7.804 | 6.684 | 0.5s |
| **VAREdit** | 2.2B | 0.271 | 0.099 | 6.210 | 7.055 | 5.662 | 0.260 | 0.225 | 7.530 | 8.083 | 6.996 | 0.7s |
| **VAREdit** | 8.4B | **0.282** | **0.125** | **8.284** | 8.461 | **7.892** | **0.266** | **0.233** | **8.621** | 8.536 | **8.105** | 1.2s |
| Step1X-Edit | 21B | 0.280 | 0.126 | 7.799 | 7.843 | 7.081 | 0.254 | 0.226 | 7.919 | 8.069 | 7.351 | 12.8s |
| FLUX.1 Kontext | 12B | 0.284 | 0.125 | 7.419 | 8.800 | 7.210 | 0.256 | 0.225 | 7.236 | 8.923 | 6.998 | 28.5s |
| Qwen-Image-Edit | 20B | 0.286 | 0.133 | 9.113 | 8.598 | 8.550 | 0.263 | 0.232 | 9.111 | 8.789 | 8.567 | 66.4s |
| GPT-4o-Image | N/A | 0.286 | 0.117 | 9.517 | 9.048 | 9.142 | 0.266 | 0.237 | 9.440 | 8.866 | 8.902 | N/A |

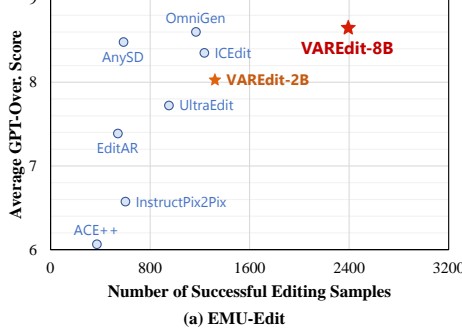
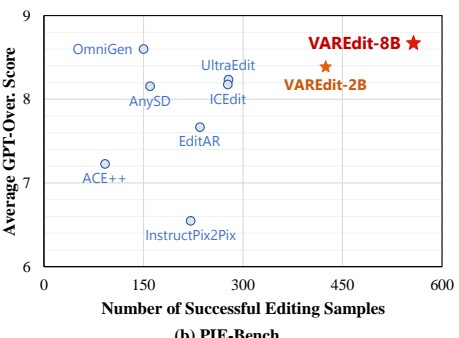

(a) EMU-Edit

(b) PIE-Bench

Figure 4: Scatter plots of successful editing samples capacity versus the GPT-Over. scores over the successful editing subsets of basic approaches on EMU-Edit and PIE-Bench benchmarks.

Notably, GPT-Bal. serves as a crucial metric since it penalizes a common failure mode: models like OmniGen can achieve high GPT-Over. scores simply by being too conservative and failing to perform the requested edit. To distinguish true preservation from mere inaction, we conduct a more rigorous analysis. We isolate only the most successful edits (GPT-Suc. ≥ 9) and re-evaluate GPT-Over. scores on the subsets for all basic approaches. Figure 4 presents the comparisons with two observations. First, VAREdit produces a significantly larger volume of high-quality edits, demonstrating the superior instruction-following capability. Second, and more critically, VAREdit achieves the highest GPT-Over. score within this successful subset, surpassing even OmniGen. This demonstrates VAREdit's ability to preserve unchanged regions.

**Robustness Across Categories**. We present the radar charts in Figure 5 that break down performance by editing types in EMU-Edit and PIE-Bench. VAREdit achieves state-of-the-art performance across the vast majority of categories among all basic approaches. While VAREdit-2B shows some limitations in challenging global style and text editing tasks, VAREdit-8B substantially closes this performance gap and beats all competitors. This illustrates the excellent scaling properties of our framework, suggesting that performance can be further enhanced by scaling to larger models and datasets. Detailed numerical results are presented in Appendix A.8.

We also present the fine-grained results in ImgEdit-Bench involving all categories in the basic-edit suite and understanding-grounding-editing (UGE) suites. According to the results in Table 2, GPT-4o-Image performs perfectly across almost all categories. Among open-source models, VAREdit-8B performs the best in background change, style transfer, removal, replacement, addition, motion change, hybrid edit and fine-grained UGE. VAREdit achieves satisfactory performance close to or even better than closed-source GPT-4o-Image, demonstrating its superiority across categories.

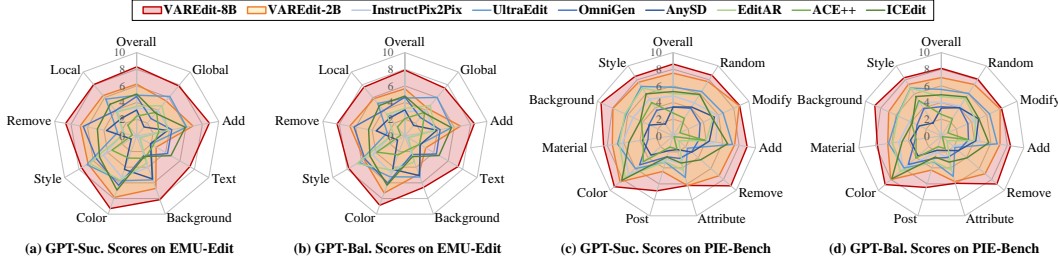

Figure 5: Fine-grained categorical GPT-Suc. and GPT-Bal. scores of VAREdit and basic approaches on EMU-Edit and PIE-Bench benchmarks.

Table 2: Quantitative results of VAREdit and representative approaches on ImgEdit-Bench.

| Method | Background | Extract | Attribute | Style | Remove | Replace | Add | Motion | Hybrid | UGE |
|---|---|---|---|---|---|---|---|---|---|---|
| UltraEdit | 3.31 | 2.02 | 3.01 | 3.69 | 1.71 | 3.13 | 3.63 | 3.57 | 2.33 | 2.36 |
| Step1X-Edit | 3.19 | 1.87 | 3.13 | 4.44 | 2.61 | 3.45 | 3.90 | 3.43 | 2.52 | 3.11 |
| GPT-4o-Image | **4.62** | **2.96** | **4.26** | **4.75** | 3.81 | 4.49 | **4.65** | **4.76** | **4.54** | **4.70** |
| VAREdit-2B | 2.95 | 1.94 | 3.33 | 4.05 | 3.51 | 3.97 | 3.57 | 2.88 | 2.57 | 2.81 |
| **VAREdit-8B** | 3.93 | 1.52 | 3.98 | 4.60 | **4.17** | **4.54** | 4.06 | 4.31 | 3.30 | 4.09 |

**Inference Efficiency**. Table 1 also demonstrates that VAREdit offers substantial efficiency improvements. VAREdit-8B completes a $512 \times 512$ edit within 1.2 seconds, making it $2.2\times$ faster than the similarly-sized UltraEdit (7.7B, 2.6s) and $7\times$ faster than the larger ICEdit model (17.0B, 8.4s). This efficiency stems from the single-pass, multi-scale generative process. Moreover, VAREdit-2B achieves a 0.7s inference time while surpassing all baselines in editing quality. While frontier editing approaches achieve impressive results, they incur a substantial computational burden, with inference times increasing by over an order of magnitude (more than $10\times$) on local hardware. This trade-off suggests that significant performance gains are attainable for VAREdit by strategically scaling its training dataset and model size, all while maintaining a moderate inference overhead.

## 4.3 QUALITATIVE RESULTS

Figure 6 provides a visual comparison that reveals the underlying reasons for VAREdit's quantitative success. In the first example, diffusion-based methods tend to overedit the image and achieve lower GPT-Over. scores. For instance, InstructPix2Pix alters the color of the entire ground and ICEdit erroneously removes the pole. The vanilla AR-based EditAR fails to execute the instruction at all, with a high GPT-Over. score yet a very low GPT-Suc. score. VAREdit successfully completes the task while precisely preserving the unchanged regions, thus achieving the highest GPT-Bal. score. Similar observations hold for the subsequent examples, confirming the effectiveness of VAREdit. To access further qualitative results and analysis, please refer to Appendix A.5.

## 4.4 ABLATIONS AND ANALYSIS

To isolate the contribution of the SAR module, we conduct an ablation study comparing the three conditioning strategies for the VAR Transformer: (1) **Full**: Conditions on features from all source scales, (2) **Finest**: Conditions on the finest-scale source features, and (3) **SAR**: Our proposed SAR-enhanced conditions (on the finest-scale source features).

Table 3: Ablation quantitative results of VAREdit-2B in full-scale, finest-scale and SAR-enhanced settings.

| Dataset | Setting | GPT Score | | | Input Length | Time |
|---|---|---|---|---|---|---|
| | | Suc. ↑ | Over. ↑ | Bal. ↑ | | |
| EMU-Edit | Full | 5.775 | 5.892 | 4.967 | 1,042 | 0.8s |
| | Finest | 5.819 | 6.480 | 5.248 | 777 | 0.5s |
| | _w/_ SAR | **6.072** | **7.058** | **5.565** | 777 | 0.5s |
| PIE-Bench | Full | 7.184 | 7.170 | 6.423 | 1,042 | 0.8s |
| | Finest | **7.291** | 7.501 | 6.588 | 777 | 0.5s |
| | _w/_ SAR | 7.234 | **7.804** | **6.684** | 777 | 0.5s |

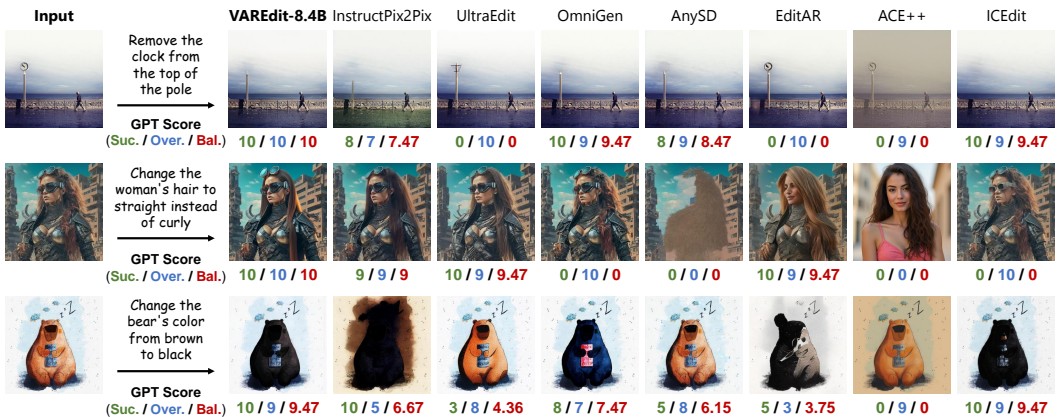

Figure 6: Qualitative results among the compared methods and VAREdit of three editing samples in EMU-Edit and PIE-Bench datasets. GPT scores are also reported to evaluate the editing quality.

The results in Table 3 and Figure 7 confirm our hypothesis on $256 \times 256$ resolution VAREdit-2B. The Full setting yields the lowest GPT-Bal. score, which can be attributed to significantly lower GPT-Over. scores. Introducing all source scales to the conditioning creates distractions for predicting target features and leads to over-editing. Moreover, this setting is 60% slower than the other two variants due to longer token sequences. In contrast, the Finest setting requires fewer tokens as input than the Full set-

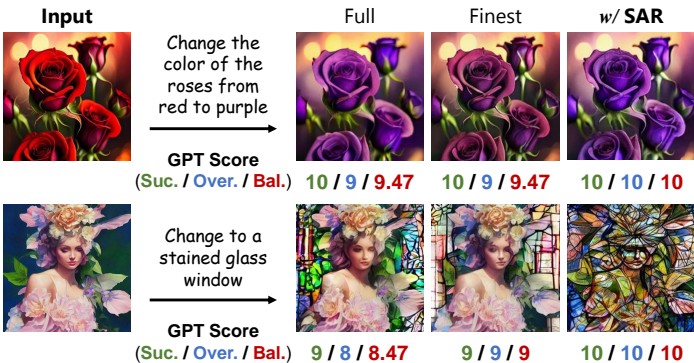

Figure 7: Ablation qualitative results of VAREdit in full-scale, finest-scale and SAR-enhanced settings.

ting, yet it suffers from scale mismatch as analyzed in Section 3.2.2 and results in poor performance. Compared with the Finest setting, the SAR-enhanced variant achieves higher GPT-Over. scores, demonstrating the effectiveness of scale-matched information injection. According to the visual results, our SAR-based variant further reduces unintended textual detail changes and incomplete style references, achieving more precise editing results.

## 5 CONCLUSION

We have proposed VAREdit, an instruction-guided image editing framework following the next-scale prediction paradigm. VAREdit takes the instruction and quantized visual token features into a VAR Transformer model to predict multi-scale residuals of the desired target image. We have analyzed the efficacy of different conditioning strategies and proposed a novel SAR module to effectively inject scale-matched conditions into the first self-attention layer. Extensive experiments have demonstrated that VAREdit achieves significantly higher editing precision scores and faster generation speed compared to state-of-the-art methods. We hope this initial exploratory study provides valuable insights into the future design of more effective and efficient AR-based visual models.

**Acknowledgement.** This work was supported by the Key Science & Technology Project of Anhui Province No. 202523o09050002, the National Natural Science Foundation of China (No. 62337001), the Key Technologies R & D Program of Anhui Province (No. 202423k09020039) and the Fundamental Research Funds for the Central Universities.

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

# A APPENDIX

## A.1 TRAINING DATASET PROCESSING DETAILS

Our training dataset for VAREdit-2B is constructed from the SEED-Data-Edit (Ge et al., 2024) and ImgEdit (Ye et al., 2025) datasets, yielding a final collection of 3.92 million paired samples. We first extract all single-turn samples of the two datasets and decompose multi-turn conversations in SEED-Data-Edit into single-turn editing pairs for the collection. We identify a significant number of noisy samples within the initial SEED-Data-Edit, characterized by poor instruction alignment, visual blurring or abnormal distortion. To address this, we filter these low-quality samples using a qualified open-source visual language model (VLM), Kimi-VL-A3B-Thinking (Team et al., 2025). The filtering prompt is shown in Figure 8. We discard approximately 1 million samples for which the VLM provided a negative judgment, resulting in our final refined dataset. For VAREdit-8B, the training dataset includes a large-scale proprietary editing samples.

> Please check whether the second image is edited based on the instruction '<INSTRUCTION>' over the first image, in good quality. Specifically:
> 1. Summarize the difference of the second image over the first image, then compare the difference and the given instruction. Do they match precisely without additional elements?
> 2. Are the two images are both clear without blurring and abnormal distortion?
> If the above aspects are both satisfied, generate "yes", otherwise generate "no", and no other content.

Figure 8: Prompt for filtering SEED-Edit-Data samples.

## A.2 ADDITIONAL DETAILS OF EVALUATION METRICS

### A.2.1 CLIP-BASED EVALUATION

For conventional CLIP-based metrics, we adhere to the established configurations of previous approaches. On the EMU-Edit (Sheynin et al., 2024) benchmark, we follow UltraEdit (Zhao et al., 2024a) to use `openai/clip-vit-base-patch32` as the base model. On the PIE-Bench (Ju et al., 2023) benchmark, we use `openai/clip-vit-large-patch14` as the base model, which is explicitly specified in the benchmark.

### A.2.2 GPT-BASED EVALUATION

Conventional metrics often fail to capture important editing quality aspects, such as spurious modifications or incomplete edits. To address this, we adopt the evaluation protocol from OmniEdit (Wei et al., 2025) to use GPT-4o (Hurst et al., 2024) as an automated judge to provide GPT-Success (Suc.) and GPT-Overedit (Over.) scores. The evaluation prompt is shown in Figure 9. We also introduce GPT-Balance (Bal.) as a balanced evaluation metric for overall editing performance, which stands for the harmonic mean of GPT-Suc. and GPT-Over. metrics. The GPT-Bal. score of an editing sample $\mathbf{e}$ is calculated as: $\text{GPT-Bal.}(\mathbf{e}) = \frac{2 \times \text{GPT-Suc.}(\mathbf{e}) \times \text{GPT-Over.}(\mathbf{e})}{\text{GPT-Suc.}(\mathbf{e}) + \text{GPT-Over.}(\mathbf{e})}$, and it takes zero value if either the edit is absolutely unsuccessful ($\text{GPT-Suc.}(\mathbf{e}) = 0$) or severe overediting occurs ($\text{GPT-Over.}(\mathbf{e}) = 0$). The final reported score for each GPT-based metric on a dataset is the arithmetic mean of the scores across all samples.

### A.2.3 ADDITIONAL EVALUATION DETAILS

For ImgEdit-Bench, we follow the official protocol that uses GPT-4o to assign 1-5 ratings, averaged across the dimensions on instruction adherence, image-editing quality, and detail preservation. For GEdit-Bench, we also use the official protocol that utilizes GPT-4.1 to assign 0-10 ratings on semantic consistency (SC), perceptual quality (PQ) and finally calculate the overall score (O) accordingly.

## A.3 DETAILS FOR COMPARED APPROACHES

We assess VAREdit against several basic tuning-based methods:

Human:
You are a professional digital artist. You will have to evaluate the effectiveness of the AI-generated image(s) based on the given rules. You will have to give your output in this way (Keep your reasoning concise and short.):{{"score": [...], "reasoning" : "..."}} and don't output anything else.
Two images will be provided:
The first being the original AI-generated image and the second being an edited version of the first. The objective is to evaluate how successfully the editing instruction has been executed in the second image.
Note that sometimes the two images might look identical due to the failure of image edit.
From a scale 0 to 10:
*A score from 0 to 10 will be given based on the success of the editing.*
- 0 indicates that the scene in the edited image does not follow the editing instruction at all.
- 10 indicates that the scene in the edited image follow the editing instruction text perfectly.
- If the object in the instruction is not present in the original image at all, the score will be 0.
*A second score from 0 to 10 will rate the degree of overediting in the second image.*
- 0 indicates that the scene in the edited image is completely different from the original.
- 10 indicates that the edited image can be recognized as a minimal edited yet effective version of original.
Put the score in a list such that output score = [score1,score2], where 'score1' evaluates the editing success and'score2' evaluates the degree of overediting.
Editing instruction: <EDITING INSTRUCTION>
<Image> Source Image </Image>
<Image> Edited Image </Image>
Assistant:

Figure 9: Prompt for evaluating the GPT-Suc. and GPT-Over. scores of the edits.

- **InstructPix2Pix** (Brooks et al., 2023) is the seminal study that introduces the natural, humanized instruction-guided image editing task. It constructs considerable paired editing samples to train an end-to-end editing model based on SD v1.5, which establishes a powerful channel-wise conditional concatenation paradigm in instruction-guided image editing.

- **UltraEdit** (Zhao et al., 2024a) introduces a large-scale dataset covering broader instruction types, real images and region editing. Canonical diffusion models trained on the UltraEdit set new benchmarking records. We adopt the published SD v3 variant for comparison.

- **OmniGen** (Xiao et al., 2025) is a diffusion-based architecture that tokenizes texts and images into token embeddings and denoises the noise tokens to reconstruct latent representations through Transformer-based rectified flow optimization. It supports unified image generation, including image editing, subject-driven and visual-conditional generation.

- **AnySD** (Yu et al., 2025) is another diffusion-based architecture integrating a task-aware routing strategy with mixture-of-experts (MoE) blocks to meet task-specific editing requirements. This model is trained based on SD v1.5 with a newly proposed comprehensive image editing dataset, AnyEdit.

- **EditAR** (Mu et al., 2025) is an AR-based architecture including a VQ-autoencoder and a token-by-token autoregressive model to predict target image tokens in latent space. To inject general visual knowledge, EditAR introduces a distillation loss from a vision foundation model as a regularization term, thereby improving the text-to-image alignment.

- **ACE++** (Mao et al., 2025) is an instruction-based diffusion framework, which adapts the long-context condition unit module to scenarios of multiple conditions. This model first experiences a zero-reference training stage and then an N-reference training stage to enable support for general instructions across various tasks.

- **ICEdit** (Zhang et al., 2025b) leverages large-scale Diffusion Transformers to construct an in-context editing framework. It integrates a LoRA-MoE hybrid tuning strategy to enhance flexible adaptation for specialized condition feature processing, while a VLM-based early filtering module is utilized to identify good samples at the very first denoising steps.

In addition, we also include four frontier image editing approaches:

- **Step1X-Edit** (Liu et al., 2025) is a unified image editing model that combines the robust semantic reasoning of MLLM with a DiT-style diffusion architecture. It maintains a good balance between conditioned image reconstruction and editing instruction following.
- **FLUX.1 Kontext** (Labs et al., 2025) is a generative flow matching approach that handles both local editing and generative in-context tasks within a DiT-style architecture.
- **Qwen-Image-Edit** (Wu et al., 2025a) is built upon Qwen-Image, simultaneously feeding the input image into Qwen2.5-VL and the VAE Encoder to obtain the representation, serving as conditions of a double-stream MMDiT architecture for effective editing.
- **GPT-4o-Image** (Hurst et al., 2024) is a closed-sourced visual generation model that supports generating highly realistic images that accurately follow instructions through simple text descriptions or image uploads.

All the compared approaches (except for GPT-4o-Image) are implemented using the officially released open-source codes with their published default parameter settings.

### A.4 ADDITIONAL IMPLEMENTATION DETAILS

Our VAREdit models are built on a foundational VAR-based model, Infinity (Han et al., 2025). We initialize our models with the released pre-trained 2B and 8B checkpoints for full fine-tuning. For visual tokenization, we use the pre-trained multi-scale tokenizers integrating binary spherical quantization (Zhao et al., 2024b), a lookup-free vector quantization method following:

$$\mathcal{Q}_{BSQ}(\mathbf{z}) = [q(z_1); q(z_2); \ldots; q(z_d)], \quad q(z_i) = \frac{1}{\sqrt{d}} \cdot \mathbf{1}_{[z_i \geq 0]}. \tag{8}$$

We set the codebook dimensions $d$ as 32 and 56 for the 2B and 8B models, respectively. During the training stages of $256 \times 256$ and $512 \times 512$ resolution, all the source and target images will be tokenized into 7 scales and 10 scales with the following lists of grid sizes $(h_k, w_k)$, respectively:

- 256px: $(1,1), (2,2), (4,4), (6,6), (8,8), (12,12), (16,16)$.
- 512px: $(1,1), (2,2), (4,4), (6,6), (8,8), (12,12), (16,16), (20,20), (24,24), (32,32)$.

In particular, we only extract the finest-scale tokens of the source images in the finest-scale conditioning strategy. We employ 2D-RoPE (Heo et al., 2024) to encode the spatial information of all input tokens before each self-attention module. For a target token at position $(i, j)$ in scale $k$, its 2D-RoPE coordinate $(x, y)$ is calculated as:

$$(x, y) = (i * \lfloor h/h_k \rfloor, j * \lfloor w/w_k \rfloor), \tag{9}$$

while the source token is treated as a spatially separate one with an offset $\Delta = (64, 64)$, indicating that its 2D-RoPE coordinate starts from $(64, 64)$:

$$(x, y) = (i * \lfloor h/h_k \rfloor + 64, j * \lfloor w/w_k \rfloor + 64). \tag{10}$$

Additionally, we also use the pre-trained scale positional embeddings in the Infinity model to mitigate inter-scale confusion.

### A.5 QUALITATIVE STUDY AND ANALYSIS

#### A.5.1 COMMON-TYPE IMAGE EDITING

To further present the superiority of VAREdit, we conduct additional qualitative analysis across diverse editing scenarios, including object-level modifications (addition, replacement, removal), attribute changes (material, text, posture, style, color) and complex compositional edits. As illustrated in Figure 10, VAREdit-8B exhibits satisfactory editing adherence to instructions and high image fidelity on these samples, which serve as commonly seen editing types.

#### A.5.2 IMAGINATIVE IMAGE EDITING

As VAREdit exhibits promising performance in common-type editing, we further explore some sophisticated editing types, including some unseen or imaginative instructions. We have presented

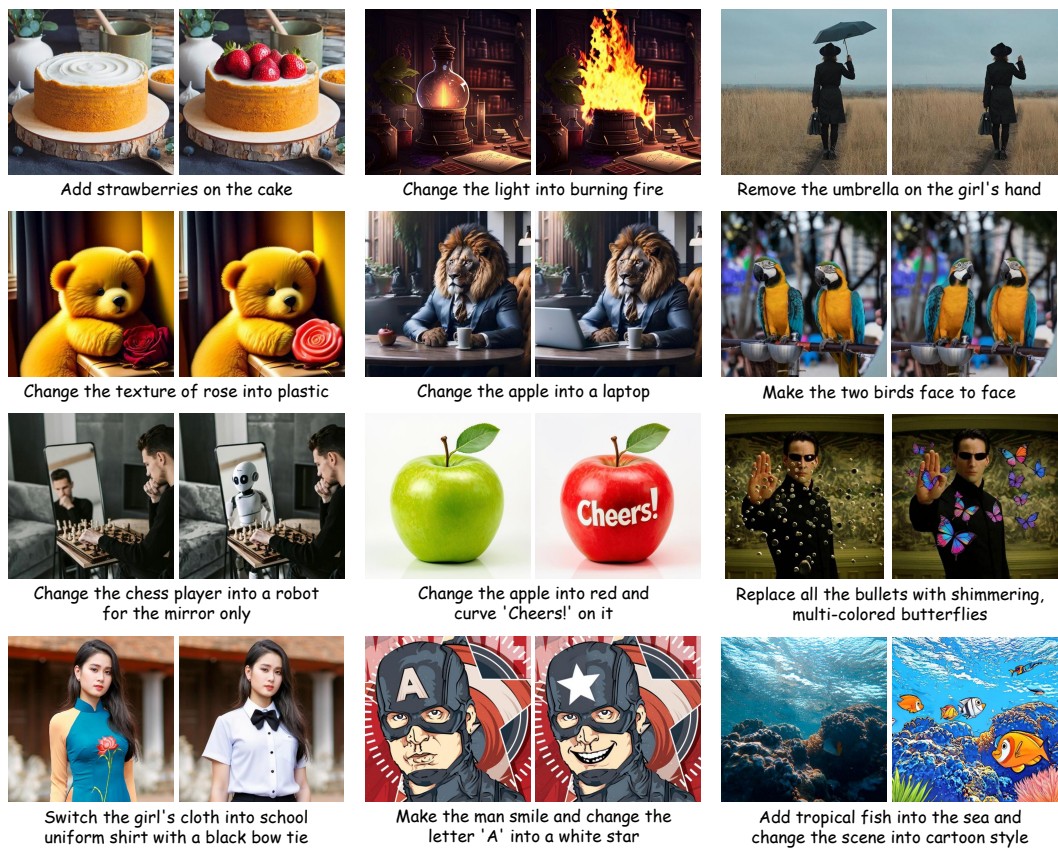

Figure 10: Qualitative editing samples of VAREdit-8B across numerous common editing types.

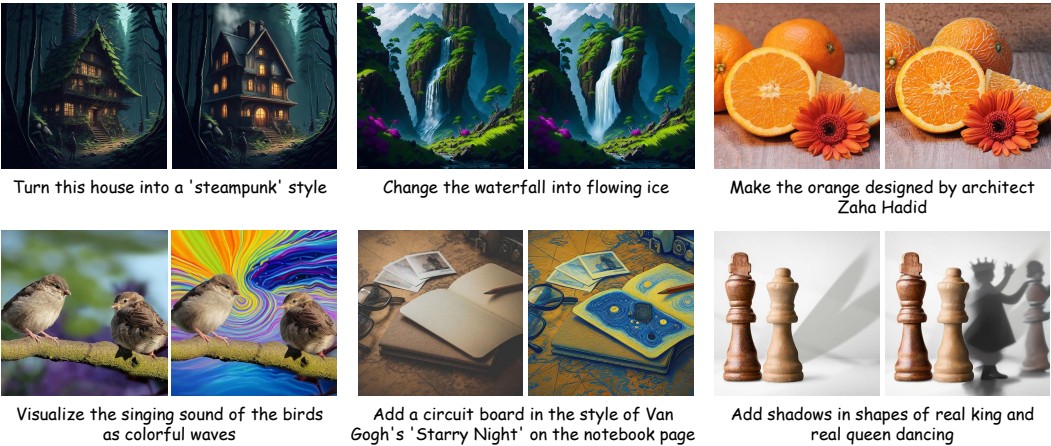

Figure 11: Qualitative editing samples of VAREdit-8B in imaginative image editing.

the results in Figure 11. Our model achieves relatively successful results on these challenging edits of artistic style transfer (cyberpunk, Zaha's design style, and Vango's starry night) and semantic conceptual synthesis (ice flowing waterfall, artistic sound waves, and chess shadows). These results strongly suggest that our model is not merely memorizing training examples but has developed generalizable editing capabilities.

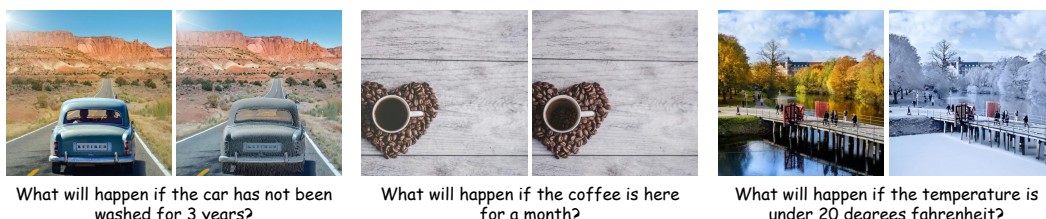

Figure 12: Qualitative editing samples of VAREdit-8B in reasoning-based image editing.

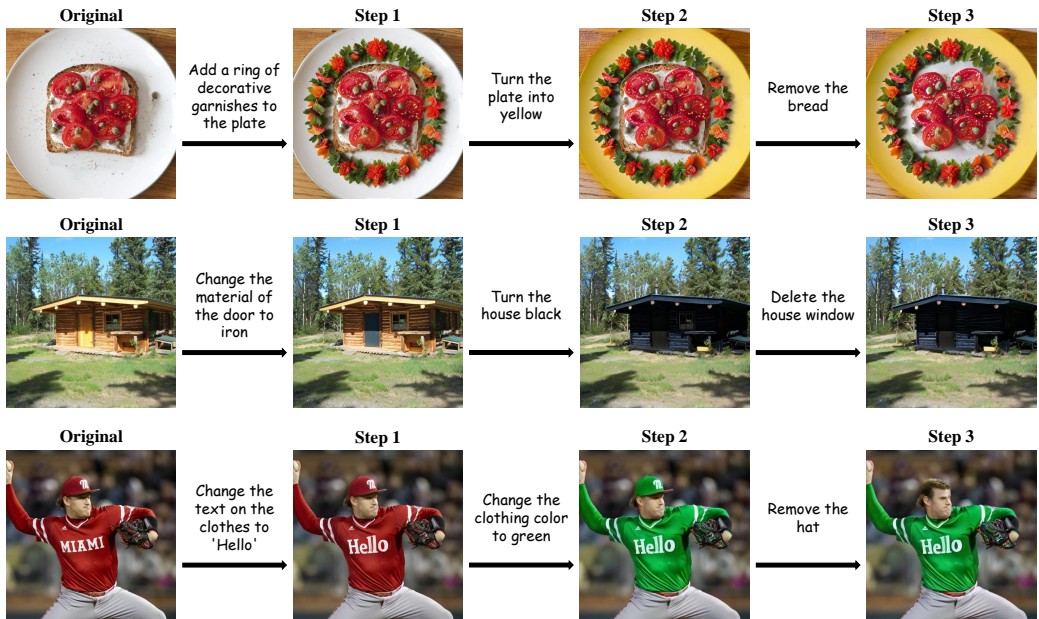

Figure 13: Qualitative samples of VAREdit-8B in multi-turn image editing.

### A.5.3 REASONING-BASED IMAGE EDITING

To evaluate the capability of VAREdit in reasoning-based image editing He et al. (2025), we present editing results of three representative image samples in ImgEdit-Bench with implicit hypothetical instructions that demand reasoning. As illustrated in Figure 12, by presenting the instruction with implicit "dirty car", "evaporation" and "cold weather", VAREdit-8B catches these points and includes the semantics in the edited samples. While some details can be further improved ("blank cup", "icy lake"), a more balanced dataset curation will better address this gap.

### A.5.4 MULTI-TURN IMAGE EDITING

A more challenging scenario is multi-turn image editing, which demands the model to preserve critical details in multiple editing steps. To address this, we present editing results of three representative image samples in ImgEdit-Bench, each with three continual editing instructions. As illustrated in Figure 13, VAREdit-8B performs well and maintains the overall quality after three editing steps.

### A.5.5 HIGHLY-UNCERTAIN IMAGE EDITING

In some scenarios, users may present unclear instructions with ambiguity or even conflicts. To investigate this, we conducted a qualitative study on several representative samples. As shown in Figure 14, unclear concepts of "look better" and "add some style" trigger no editing effect, while the underspecified "make the sky louder" is executed by removing the cloud unexpectedly. For con-

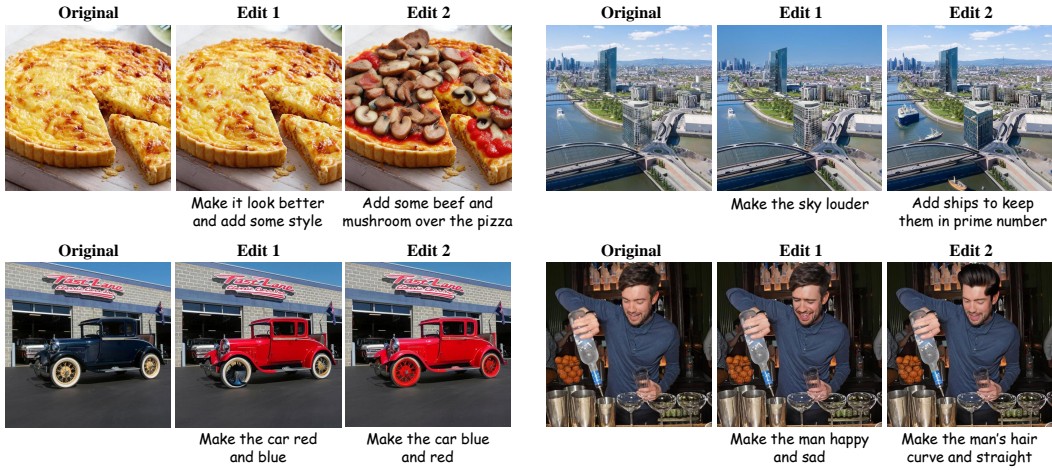

Figure 14: Additional editing samples of VAREdit-8B in highly-uncertain image editing.

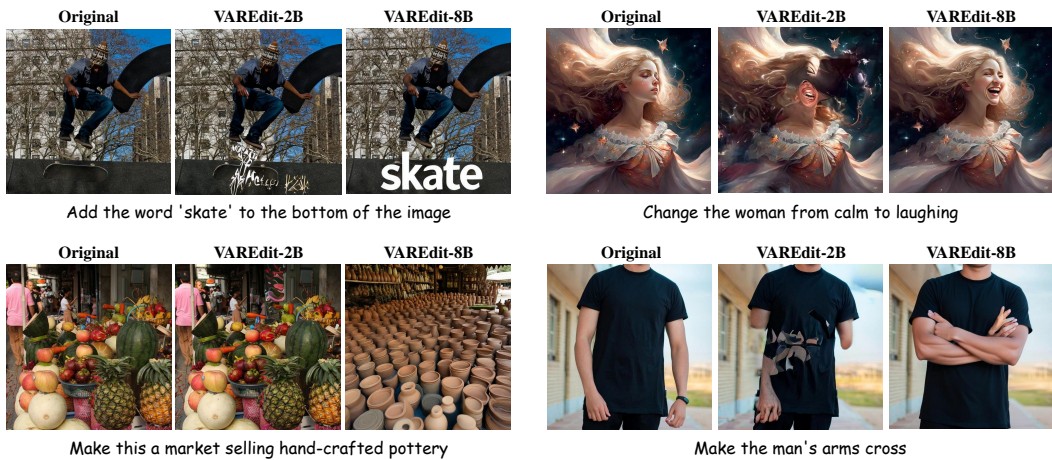

Figure 15: Failure editing case results of VAREdit-2B and VAREdit-8B.

flicting instructions, VAREdit exhibits different patterns. By switching "red" and "blue", VAREdit tends to ignore "blue" rather than considering both, illustrating a biased focus in some elements. In contrast, the "happy-and-sad" and "curve-and-straight" have been executed at a somewhat balanced level rather than ignoring one of them. In summary, VAREdit demonstrates a degree of robustness in the face of less-than-ideal instructions, and its behavior patterns (converging to priors, ignoring noise, or compromising in conflict) are predictable.

### A.5.6 FAILURE CASE ANALYSIS

To comprehensively explore the limitations of VAREdit, we conduct failure case analysis on both 2B and 8B sizes and present some representative editing results. As illustrated in Figure 15, our investigation reveals that while VAREdit-2B struggles with specific categories (*e.g.*, text generation, motion change), VAREdit-8B largely mitigates these issues through scaling. However, even the larger model encounters difficulties in complex compositional tasks that require global modification and preservation of multiple distinct local elements. For example, VAREdit-8B modifies the market into pottery successfully, yet the shapes of some pottery remain abnormal. Moreover, the cross-arm edit poses difficulty to the 8B model by maintaining the general hand fingers. We attribute this limitation to the current density of processing scales and hypothesize that expanding both the image resolution and training data will resolve these fine-grained failures.

### A.5.7 USER CROSS-VALIDATE STUDY

While we use CLIP and GPT-based scores as the main evaluation metrics, it is unknown how well these values reflect the user preferences. To mitigate this gap, we conducted a blind, randomized human preference study on the PIE-Bench dataset. For each sample in PIE-Bench, we presented participants with the source image, the text instruction, and the three edited images generated by our model (VAREdit-8B) and two strong baselines (Step1X-Edit, FLUX.1 Kontext (dev)). The three generated images were displayed in a randomized order to avoid positional bias. Participants were asked to select the single best image that satisfies two criteria: (1) Instruction following: how accurately the image reflects the textual instruction, and (2) Image quality: the realism of the edit and the absence of artifacts or over-editing. We collected responses from 5 participants (e.g., graduate students in our field with experience in evaluating generative models). The consistency in the results is expected to provide a strong and valuable signal regarding user preference.

We report both the average number of wins and the overall preference rate for each model. As illustrated in Table 4, the three methods received a globally balanced number of wins, while VAREdit-8B achieved the highest preference rate (36.0%). This study confirms that VAREdit's quantitative strengths translate directly into superior perceptual quality for users.

Table 4: Cross-validate user study results on PIE-Bench. The best group results are marked in **bold**.

| Method | Avg. Count | Avg. Rate (%) | GPT-Bal. |
|---|---|---|---|
| Step1X-Edit | 210.4 | 30.1 | 7.351 |
| FLUX.1 Kontext (dev) | 237.4 | 33.9 | 6.998 |
| **VAREdit-8B (Ours)** | **252.2** | **36.0** | **8.106** |

### A.6 QUANTITATIVE RESULTS ON GEDIT-BENCH

We present quantitative results on GEdit-Bench, reporting the SC score, PQ score and the overall score evaluated by GPT-4.1 on VAREdit and some representative approaches. As illustrated in Table 5, GPT-4o-Image scores the highest, while Step1X-Edit and VAREdit-8B are the leading open-source methods. While the absence of training data with Chinese instructions may influence the editing performance of VAREdit, an improved scaling and balance of the training hours and data volume can mitigate this gap.

Table 5: Quantitative results of VAREdit and representative approaches on GEdit-Bench (English).

| Method | Full | | | Intersection | | |
|---|---|---|---|---|---|---|
| | G_SC | G_PQ | G_O | G_SC | G_PQ | G_O |
| InstructPix2Pix | 3.335 | 6.210 | 3.234 | 3.296 | 6.189 | 3.219 |
| AnySD | 3.122 | 5.865 | 2.919 | 3.053 | 5.882 | 2.854 |
| OmniGen | 6.037 | 5.856 | 5.154 | 5.879 | 5.871 | 5.005 |
| Step1X-Edit | 7.289 | 6.962 | 6.618 | 7.131 | 6.998 | 6.444 |
| GPT-4o-Image | **7.867** | **8.097** | **7.590** | **7.743** | **8.133** | **7.494** |
| **VAREdit-2B** | 5.121 | 5.301 | 4.460 | 4.992 | 5.220 | 4.334 |
| **VAREdit-8B** | 7.292 | 6.581 | 6.383 | 7.092 | 6.626 | 6.236 |

### A.7 SELF-ATTENTION HEATMAPS ANALYSIS

We provide the full-layer self-attention heatmaps in Figure 16 as a more comprehensive scale dependency analysis. It can be seen that in the first attention layer, the attention scores distribute broadly on almost all source scales, especially the coarser source scales. While in other layers, the attention patterns become highly localized. Specifically, attention scores between source scales and target scales are in strong diagonal structures in deeper Transformer layers, suggesting that attention is primarily confined to tokens in the spatial neighborhood. In this task, the finest accumulated source scale $\mathbf{F}_K^{(src)}$ is sufficient to provide reference information for high-quality image editing. Such observations inspire us to design the SAR module to inject scale-matched conditions into the first self-attention layer, while maintaining the finest-scale conditional setting in deeper layers.

To investigate the attention pattern of the SAR variant, we visualize self-attention heatmaps from some intermediate layers. As illustrated in Figure 17, the first layer exhibits a clear scale-aligned behavior. The attention scores $(\mathbf{Q}_k^{(tgt)}\mathbf{K}_k^{(ref)\top}/\sqrt{d})$ assigned by each target scale $k$ to the corresponding reference features are predominantly concentrated, without focus on the finest-scale source features. In subsequent layers, the attention pattern becomes highly localized as the same in the full-

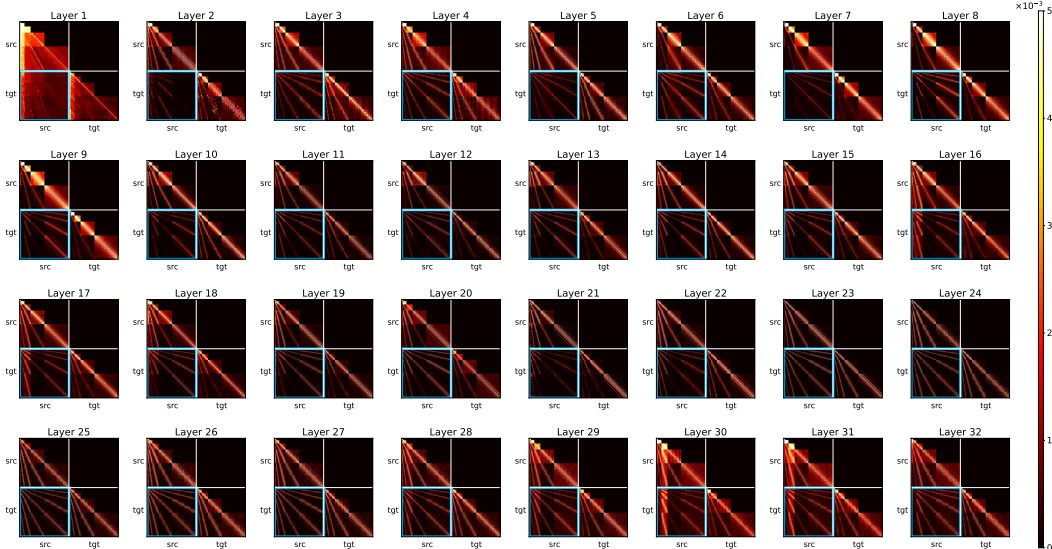

Figure 16: Self-attention heatmaps based on the full-scale setting among all transformer layers.

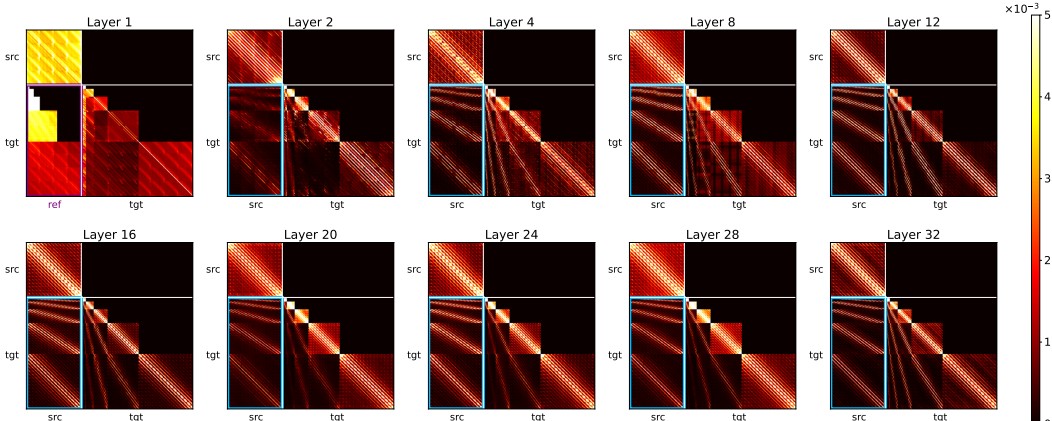

Figure 17: Self-attention heatmaps of PIE-Bench samples based on the SAR-variant across different layers. Note that in the first layer, the focused zone is organized not strictly aligned with the corresponding positions, since $\mathbf{F}_k^{(ref)}$ has a spatial size of $(h_k, w_k)$ instead of the largest scale $(h_K, w_K)$.

scale setting. This behavior mirrors the attention patterns observed in the standard full-scale setting, suggesting a shift from global scale-matching to local refinement.

## A.8 NUMERICAL RESULTS FOR CATEGORICAL EVALUATION

Table 6, 7, 8, 9 present the detailed numerical results of the fine-grained categorical scores shown in the radar charts of the main context. It can be observed that VAREdit-2B outperforms the compared approaches across the vast majority of editing types, such as addition, removal, coloring and material transfer. VAREdit-8B enjoys enhanced performance by a substantial margin that outperforms all competitors. According to these tables, the 2B model shows some limitations in challenging scenarios, including global editing and text editing. Through scaling up the model size and training iterations, the 8B model substantially mitigates these gaps, especially in text editing that achieves 150%+ improvements over the 2B variant. These findings strongly demonstrate that our VAREdit is not only effective but also highly scalable. As a result, our VAREdit models exhibit high robustness across various editing types to achieve precise editing results.

Table 6: Numerical fine-grained categorical GPT-Suc. scores of all methods on EMU-Edit. Among column names, "Back." stands for "Background". The best group results are marked in **bold**, while the second-best group results are underlined.

| Method | Overall | Global | Add | Text | Back. | Color | Style | Remove | Local |
|---|---|---|---|---|---|---|---|---|---|
| InstructPix2Pix | 3.358 | 3.950 | 4.034 | 0.330 | 4.043 | 5.270 | 4.634 | 1.637 | 3.897 |
| UltraEdit | 4.881 | 6.178 | 6.084 | 1.985 | 5.625 | 5.728 | 6.889 | 2.328 | 5.753 |
| OmniGen | 4.738 | 2.703 | 4.274 | 4.268 | 2.509 | 6.324 | 5.558 | 6.516 | 3.960 |
| AnySD | 3.098 | 1.379 | 3.752 | 1.899 | 5.501 | 4.289 | 0.744 | 3.684 | 2.789 |
| EditAR | 3.582 | 4.717 | 3.797 | 0.472 | 3.488 | 5.678 | 6.571 | 1.501 | 3.697 |
| ACE++ | 2.375 | 3.717 | 2.694 | 2.288 | 2.871 | 2.832 | 3.438 | 0.535 | 1.650 |
| ICEdit | 5.027 | 3.548 | 5.931 | 4.751 | 2.635 | 6.935 | 5.138 | 4.793 | 4.955 |
| **VAREdit-2B** | 6.210 | 5.475 | 6.865 | 2.708 | 6.724 | 7.854 | 6.740 | 6.889 | 6.314 |
| **VAREdit-8B** | **8.284** | **7.827** | **8.857** | **7.375** | **8.150** | **9.285** | **7.641** | **8.657** | **8.040** |

Table 7: Numerical fine-grained categorical GPT-Bal. scores of all methods on EMU-Edit. Among column names, "Back." stands for "Background". The best group results are marked in **bold**, while the second-best group results are underlined.

| Method | Overall | Global | Add | Text | Back. | Color | Style | Remove | Local |
|---|---|---|---|---|---|---|---|---|---|
| InstructPix2Pix | 2.923 | 3.934 | 3.477 | 0.326 | 2.607 | 4.740 | 4.514 | 1.469 | 3.204 |
| UltraEdit | 4.541 | 6.020 | 5.695 | 2.071 | 5.256 | 5.221 | 6.346 | 2.159 | 5.079 |
| OmniGen | 4.666 | 2.877 | 4.293 | 4.366 | 2.390 | 6.036 | 5.539 | 6.206 | 3.979 |
| AnySD | 3.129 | 1.630 | 3.893 | 2.124 | 5.072 | 4.276 | 1.012 | 3.497 | 2.818 |
| EditAR | 3.305 | 4.753 | 3.622 | 0.472 | 3.321 | 4.392 | 6.648 | 1.381 | 3.364 |
| ACE++ | 2.076 | 3.689 | 2.070 | 2.021 | 1.660 | 2.422 | 3.899 | 0.510 | 1.392 |
| ICEdit | 4.785 | 3.603 | 5.673 | 4.732 | 2.651 | 6.301 | 4.850 | 4.427 | 4.756 |
| **VAREdit-2B** | 5.662 | 4.279 | 6.775 | 2.887 | 4.882 | 7.303 | 5.661 | 6.716 | 5.821 |
| **VAREdit-8B** | **7.892** | **7.473** | **8.460** | **7.375** | **6.573** | **8.870** | **7.788** | **8.274** | **7.652** |

Table 8: Numerical fine-grained categorical GPT-Suc. scores of all methods on PIE-Bench. Among column names, "Attr.", "Mater.", "Back." stands for "Attribute", "Material", "Background", respectively. The best group results are marked in **bold**, while the second-best group results are underlined.

| Method | Overall | Random | Modify | Add | Remove | Attr. | Post | Color | Mater. | Back. | Style |
|---|---|---|---|---|---|---|---|---|---|---|---|
| InstructPix2Pix | 4.794 | 4.707 | 5.913 | 5.163 | 2.313 | 3.425 | 2.950 | 7.075 | 6.175 | 5.275 | 5.238 |
| UltraEdit | 5.831 | 6.336 | 6.588 | 7.200 | 2.275 | 5.200 | 3.300 | 6.000 | 6.750 | 6.200 | 7.050 |
| OmniGen | 3.459 | 3.921 | 3.563 | 3.138 | 2.938 | 2.825 | 2.625 | 5.325 | 3.050 | 3.200 | 3.650 |
| AnySD | 3.456 | 4.114 | 5.438 | 4.388 | 2.988 | 1.725 | 1.850 | 3.525 | 3.725 | 3.113 | 1.700 |
| EditAR | 5.070 | 5.836 | 5.263 | 4.000 | 0.900 | 4.300 | 3.225 | 7.600 | 6.525 | 6.363 | 6.800 |
| ACE++ | 2.743 | 2.514 | 0.888 | 3.875 | 0.150 | 1.650 | 1.400 | 4.750 | 3.600 | 4.225 | 4.763 |
| ICEdit | 5.321 | 5.900 | 5.150 | 6.613 | 4.388 | 3.400 | 2.650 | 8.150 | 5.350 | 4.325 | 5.988 |
| **VAREdit-2B** | 7.530 | 7.743 | 8.725 | 7.463 | 7.300 | 6.225 | 4.450 | 8.425 | 7.375 | 7.950 | 7.663 |
| **VAREdit-8B** | **8.621** | **8.643** | **8.775** | **8.975** | **9.163** | **6.200** | **6.875** | **9.325** | **8.425** | **9.513** | **8.475** |

Table 9: Numerical fine-grained categorical GPT-Bal. scores of all methods on PIE-Bench. Among column names, "Attr.", "Mater.", "Back." stands for "Attribute", "Material", "Background", respectively. The best group results are marked in **bold**, while the second-best group results are underlined.

| Method | Overall | Random | Modify | Add | Remove | Attr. | Post | Color | Mater. | Back. | Style |
|---|---|---|---|---|---|---|---|---|---|---|---|
| InstructPix2Pix | 4.034 | 4.251 | 4.328 | 4.711 | 1.861 | 2.767 | 2.016 | 6.068 | 5.147 | 4.174 | 4.781 |
| UltraEdit | 5.580 | 6.056 | 6.369 | 6.775 | 2.136 | 5.066 | 3.256 | 5.898 | 6.394 | 5.807 | 6.830 |
| OmniGen | 3.498 | 4.034 | 3.507 | 3.289 | 2.708 | 2.788 | 2.788 | 5.245 | 3.170 | 3.193 | 3.888 |
| AnySD | 3.326 | 4.022 | 4.923 | 4.183 | 2.703 | 1.814 | 1.805 | 3.695 | 3.752 | 2.892 | 1.832 |
| EditAR | 4.707 | 5.312 | 4.540 | 3.966 | 0.908 | 4.279 | 3.193 | 6.527 | 5.902 | 5.725 | 6.800 |
| ACE++ | 2.574 | 2.458 | 0.853 | 3.192 | 0.151 | 1.586 | 1.341 | 4.067 | 3.047 | 3.972 | 5.032 |
| ICEdit | 4.933 | 5.539 | 4.961 | 5.756 | 4.056 | 3.289 | 2.645 | 7.866 | 4.750 | 3.814 | 5.611 |
| **VAREdit-2B** | 6.996 | 7.330 | 7.761 | 7.112 | 6.942 | 5.717 | 4.291 | 8.032 | 6.502 | 6.950 | 7.348 |
| **VAREdit-8B** | **8.105** | **8.093** | **7.940** | **8.392** | **8.831** | **5.936** | **6.453** | **8.919** | **7.755** | **8.779** | **8.287** |

