# OpenReview forum: "Visual Autoregressive Modeling for Instruction-Guided Image Editing"
_ICLR.cc/2026/Conference — ICLR 2026 Poster_

### Official Review · Reviewer_dF8r · 2025-10-22

**Soundness:** 3
**Presentation:** 3
**Contribution:** 3
**Rating:** 6
**Confidence:** 3

**Summary:**

This paper introduces VAREdit, the first tuning-based visual autoregressive model for instruction-guided image editing. It proposes the novel SAR module to mitigate the scale mismatch problem in the VAR tuning process. VAREdit achieves SOTA across different image editing benchmarks. The paper is well written and organized, despite the lack of reasoning-based and multi-step image editing analysis.

**Strengths:**

1. Novel Paradigm: The paper introduces VAREdit, the first tuning-based visual autoregressive (VAR) framework for instruction-guided image editing, which reframes editing as a next-scale prediction task. The proposed Scale-Aligned Reference (SAR) module effectively mitigates the scale mismatch problem by injecting scale-matched conditioning only into the first self-attention layer. The motivation and design are well-analyzed through self-attention heatmap studies.

2. Efficiency and Scalability: VAREdit achieves 2.2× faster inference than comparable diffusion models while maintaining superior quality. The scalability from 2.2B to 8.4B models shows consistent improvement across all editing types.

3. Well Writing and Structure: The paper is clearly structured, with detailed methodological explanations, visualizations, and ablations. It maintains excellent reproducibility and clarity in presentation

**Weaknesses:**

Because of its AR-based nature, I wonder if such a VAR-based image editing method can show strong ability in reasoning-based image editing and multi-step image editing tasks. If the author can include some relevant experiments, such as WISE or GEdit, I would consider raise the score.

**Questions:**

See the weakness

---

> ### Author Response · Authors · 2025-11-26
> **Author Rebuttal [1/2]**
>
> Dear Reviewer dF8r,
>
> Thank you very much for acknowledging the strengths of our study and your constructive feedback.
> We have carefully considered all of your suggestions that make our study more convincing.
> Here are the detailed responses:
>
> > **W1.**  VAREdit's ability in reasoning-based editing and multi-step image editing tasks are unknown. Additional results on new benchmarks (e.g., WISE, GEdit) are required.
>
> Thanks for the suggestion. We have conducted further evaluations on more comprehensive image editing benchmarks. Since WISE is a dataset of text-to-image task, we switch to evaluate VAREdit on ImgEdit-Bench and GEdit-Bench.
>
> - For ImgEdit-Bench, we present the result table involving all categories in basic-edit suite and understanding-grounding-editing (UGE) suites:
>
>     |Category|VAREdit-8B|VAREdit-2B|GPT-4o-Image|Step1X-Edit|ImgEdit-E1|UltraEdit|
>     |:--|:--:|:--:|:--:|:--:|:--:|:--:|
>     |Background Change|3.93|2.95|**4.62**|3.19|3.38|3.31|
>     |Object Extraction|1.52|1.94|**2.96**|1.87|2.55|2.02
>     |Attribute Alter|3.98|3.33|**4.26**|3.13|4.04|3.01|
>     |Style Transfer|4.60|4.05|**4.75**|4.44|4.38|3.69|
>     |Removement|**4.17**|3.51|3.81|2.61|2.40|1.71|
>     |Replacement|**4.54**|3.97|4.49|3.45|2.80|3.13|
>     |Addition|4.06|3.57|**4.65**|3.90|3.82|3.63|
>     |Motion Change|4.31|2.88|**4.76**|3.43|3.21|3.57|
>     |Hybrid Edit|3.30|2.57|**4.54**|2.52|2.87|2.33|
>     |UGE Score|4.09|2.81|**4.70**|3.11|3.20|2.36|
>
>     VAREdit-8B demonstrates robust generalization to real-world images, achieving the highest scores among open-source models in tasks like Removal, Replacement, and Motion Change. In the fine-grained UGE suit, it significantly outperforms baselines like Step1X-Edit (4.09 vs. 3.11). As a closed-source model, GPT-4o-Image achieves better performance with much larger training data and model size.
>
>
>
> - For GEdit-Bench, we first present the result tables involving all categories of VAREdit-8B and VAREdit-2 (on English instructions):
>
>     - Categorical Results on VAREdit-8B:
>
>         |Category| Intersection (G_SC) | Intersection (G_PQ) | Intersection (G_O) | Full (G_SC) | Full (G_PQ) | Full (G_O) |
>         |:--:|:--:|:--:|:--:|:--:|:--:|:--:|
>         |Background Change|7.862|5.793|6.433|7.875|5.950|6.480|
>         |Color Alter|9.000|6.265|7.320|9.050|6.600|7.557|
>         |Material Alter|7.500|5.714|6.215|7.300|5.700|6.016|
>         |Motion Change|5.864|6.591|5.249|4.725|6.550|4.067|
>         |PS Human|4.244|8.293|4.500|4.400|8.057|4.600|
>         |Style Change|7.771|5.958|6.572|7.483|5.983|6.450|
>         |Subject Add|8.237|6.842|7.400|7.917|6.933|7.132|
>         |Subject Remove|8.548|6.976|7.558|8.491|7.053|7.522|
>         |Subject Replace|8.261|6.348|6.989|8.050|6.217|6.777|
>         |Text Change|4.481|7.136|4.673|4.323|7.172|4.588|
>         |Tone Transfer|8.440|6.480|7.301|8.400|6.675|7.405|
>         |**Average**|7.292|6.581|6.383|7.092|6.626|6.236|
>
>     - Categorical Results on VAREdit-2B:
>
>         |Category| Intersection (G_SC) | Intersection (G_PQ) | Intersection (G_O) | Full (G_SC) | Full (G_PQ) | Full (G_O) |
>         |:--:|:--:|:--:|:--:|:--:|:--:|:--:|
>         |Background Change|5.793|4.897|4.749|5.925|4.700|4.832 |
>         |Color Alter|7.735|5.529|6.343|7.700|5.625|6.409 |
>         |Material Alter|4.714|4.071|3.982|4.625|4.500|3.952 |
>         |Motion Change|3.136|6.182|2.543|1.875|5.750|1.545 |
>         |PS Human|3.000|6.146|3.086|2.943|6.271|3.003 |
>         |Style Change|5.958|4.000|4.490|6.017|3.867|4.466 |
>         |Subject Add|6.500|6.079|5.800|6.267|5.783|5.616 |
>         |Subject Remove|6.857|6.333|6.290|6.632|6.281|6.132 |
>         |Subject Replace|6.500|4.696|5.135|6.450|4.767|5.191 |
>         |Text Change|1.778|5.383|2.293|1.980|5.525|2.412 |
>         |Tone Transfer|4.360|5.000|4.343|4.500|4.350|4.113 |
>         |**Average**|5.121|5.301|4.460|4.992|5.220|4.334 |
>
>     We then present the comparison results across representative approaches:
>
>     - Overall Comparison Results:
>
>         |Approach| Intersection (G_SC) | Intersection (G_PQ) | Intersection (G_O) | Full (G_SC) | Full (G_PQ) | Full (G_O) |
>         |:--:|:--:|:--:|:--:|:--:|:--:|:--:|
>         |InstructPix2Pix|3.335|6.210|3.234|3.296|6.189|3.219|
>         AnySD|3.122|5.865|2.919|3.053|5.882|2.854|
>         OmniGen|6.037|5.856|5.154|5.879|5.871|5.005|
>         Step1X-Edit|7.289|6.962|6.618|7.131|6.998|6.444|
>         GPT-4o-Image|**7.867**|**8.097**|**7.590**|**7.743**|**8.133**|**7.494**|
>         VAREdit-2B|5.121|5.301|4.460|4.992|5.220|4.334 |
>         VAREdit-8B|7.292|6.581|6.383|7.092|6.626|6.236 |
>
> Based on the fine-grained categorical results, VAREdit demonstrates exceptional advantages in Color Alteration, Subject Removal, Tone Transfer, and Subject Addition, consistently achieving high scores in both instruction adherence (G_SC) and perceptual quality (G_PQ).

---

> ### Author Response · Authors · 2025-11-26
> **Author Rebuttal [2/2]**
>
> (...**W1**) Regarding the relatively lower performance in Text Change and PS Human, we attribute this to specific gaps in data distribution. Specifically, our training set lacks the Chinese character samples present in the GEdit benchmark (e.g., transforming "大话西游" to "神话懂游") and contains limited coverage of face beautification data. We anticipate that scaling the training data to encompass these domains will effectively improve the performance.
>
> The aggregate results confirm that VAREdit-8B is highly competitive with Step1X-Edit and surpasses all other open-source baselines. It is worth noting that Step1X-Edit appears to benefit from a training distribution that is more closely aligned with the specific tasks found in GEdit-Bench.
>
> In addition, more qualitative case results have been presented in the revised manuscript (**Appendix A.5**), with more diverse editing types, e.g., reasoning-based editing (**Appendix A.5.3**), multi-turn editing (**Appendix A.5.4**).
>
> **References**
>
> [1] He Q, Chen X, Wang C, et al. Reasoning to Edit: Hypothetical Instruction-Based Image Editing with Visual Reasoning[J]. arXiv preprint arXiv:2507.01908, 2025.

---

### Official Review · Reviewer_jZVb · 2025-10-27

**Soundness:** 3
**Presentation:** 3
**Contribution:** 2
**Rating:** 4
**Confidence:** 5

**Summary:**

The paper proposes VAREdit, a visual autoregressive (VAR) framework that reframes instruction-guided image editing as a next-scale prediction problem. Given an instruction and a source image, a multi-scale tokenizer plus a VAR Transformer predicts target residuals from coarse to fine. A key observation is that using only finest-scale source features to condition coarse target predictions causes a scale mismatch. The authors introduce a Scale-Aligned Reference (SAR) module that injects scale-matched conditioning only in the first self-attention layer, while subsequent layers keep using finest-scale conditioning. On EMU-Edit and PIE-Bench, VAREdit reports higher CLIP and GPT-based metrics than diffusion and AR baselines, and achieves ~1.2 s latency for 512*512 edits with an 8.4B model.

**Strengths:**

- The paper introduces a novel method called Scale-Aligned Reference (SAR), which effectively balances computational efficiency and editing performance.
- The proposed method achieves strong performance on EMU-Edit and PIE-Bench, with moderate latency, outperforming several diffusion-based and autoregressive (AR) baselines.
- The paper includes comprehensive experimental analysis; it conducts attention-level investigations to motivate the design of SAR, provides both qualitative and quantitative results, and performs ablation studies to demonstrate the effectiveness of SAR.

**Weaknesses:**

- the paper provides self-attention heatmaps based on the full-scale setting to motivate the design of SAR, but a similar analysis of the tuned model is missing; including this will cross-validate the modeling choice and strengthen the motivation.
- the paper relies on GPT score to evaluate editing models; however, GPT judges may introduce hallucinations and prompt-induced biases. A better way would be to use a controlled human study to cross-validate the reliability of using GPT as the evaluator.
- while the method demonstrates strong performance, it does not compare against frontier editing models such as GPT-4o-Image, FLUX.1 Kontext, or Qwen-Image.
- the authors claim their model as "the first tuning-based visual autoregressive model for instruction-guided image editing.", which is false, see [NEP: Autoregressive Image Editing via Next Editing Token Prediction](https://arxiv.org/pdf/2508.06044). VAR broadly means autoregressive modeling of visuals. The authors might be using it to indicate "image generation via next scale prediction," but it is very misleading. It would be better to spell it out.

**Questions:**

Following the Weakness section, my questions are as follows:

Q1: Could you provide the self-attention heatmaps under the SAR setting? I would like to visually confirm how SAR changes the attention distribution compared to the full-scale setting.

Q2: Can you provide references to support the claim in lines 131-132: *Compared to diffusion
models, AR approaches generally offer superior prompt adherence and faster inference.*

Q3: Please can you include performance comparisons with existing frontier editing models (e.g., GPT-4o-Image, Qwen-Image, etc.) in your experiments?

---

> ### Author Response · Authors · 2025-11-26
> **Author Rebuttal [1/2]**
>
> Dear Reviewer jZVb,
>
> Thank you very much for acknowledging the strengths of our study and your constructive suggestions.
>
> > **W1.**  This paper does not provide self-attention heatmap analysis of the tuned SAR model.
>
> Thank you for the insightful suggestion. To explore the attention pattern of the SAR variant, we provide the corresponding self-attention heatmaps of some intermediate layers on PIE-Bench as the same to the full-scale setting.
> In the first layer, the attention scores assigned to the reference features are extremely significant, demonstrating an effective scale-aware alignment as the matched reference.
> To access more details, please refer to our revised manuscript (**Appendix A.6**, **Figure 17**).
>
> > **W2.**  This paper relies on GPT score to evaluate models instead of a controlled human study to cross-validate the reliability.
>
>
> Thanks for the suggestion. We fully agree that human evaluation is necessary for assessing subjective quality such as over-editing artifacts and semantic alignment. To mitigate this gap, we conducted a blind, randomized human preference study on the PIE-Bench dataset (700 samples).
>
> - **Study Design**: For each of the 700 samples in PIE-Bench, we presented participants with the source image, the text instruction, and the three edited images generated by our model (VAREdit-8B) and two strong baselines (Step1X-Edit, FLUX.1 Kontext (dev)). The three generated images were displayed in a randomized order to avoid positional bias.
>
> - **Task**: This is a blind, forced-choice study. Participants were asked to select the single best image that satisfies two criteria: (1) Instruction Following: How accurately the image reflects the textual instruction. (2) Image Quality: The realism of the edit and the absence of artifacts or over-editing.
>
> - **Participants**: We collected responses from 5 participants (e.g., graduate students in our field with experience in evaluating generative models).
> While we acknowledge this is a small-scale study due to the time constraints of the rebuttal period, the consistency in the results provides a strong and valuable signal regarding user preference.
>
> - **Results**: We report both the average number of wins and the overall preference rate for each model:
>
>     |Model|Avg. Win Count|Preference Rate (%)|GPT-Bal.|
>     |:--:|:--:|:--:|:--:|
>     |Step1X-Edit|210.4|30.1|7.351|
>     |FLUX.1 Kontext (dev)|237.4|33.9|6.998|
>     |**VAREdit-8B**|**252.2**|**36.0**|**8.106**|
>
> VAREdit-8B achieved the highest preference rate (36.0%), outperforming both strong competitors. This study confirms that VAREdit's quantitative strengths translate directly into superior perceptual quality for users. Further details are provided in our revised manuscript (**Appendix A.5.7**).
>
> > **W3.**  This model does not compare against frontier editing models such as GPT-4o-Image, FLUX.1 Kontext or Qwen-Image (Edit).
>
> Many thanks for your suggestion.
> To improve the evaluation, we have further selected four frontier image editing approaches for comparison: Step1X-Edit, GPT-4o-Image, FLUX.1 Kontext (dev) and Qwen-Image-Edit.
> We report the evaluation reports on both EMU-Edit and PIE-Bench.
>
> - Results on EMU-Edit:
>
>     |Approach|CLIP-Out.|CLIP-Dir.|GPT-Suc.|GPT-Over.|GPT-Bal.|
>     |:--:|:--:|:--:|:--:|:--:|:--:|
>     |Step1X-Edit|0.280|0.126|7.799|7.843|7.081|
>     |GPT-4o-Image|**0.286**|0.117|9.517|9.048|9.142|
>     |FLUX.1 Kontext (dev)|0.284|0.125|7.419|8.800|7.210|
>     |Qwen-Image-Edit|**0.286**|**0.133**|9.113|8.598|8.550|
>     |**VAREdit-8B**|0.282|0.125|8.284|8.461|7.892|
>
> - Results on PIE-Bench:
>
>     |Approach|CLIP-Whole|CLIP-Edit|GPT-Suc.|GPT-Over.|GPT-Bal.|
>     |:--:|:--:|:--:|:--:|:--:|:--:|
>     |Step1X-Edit|0.254|0.226|7.919|8.069|7.351|
>     |GPT-4o-Image|**0.266**|**0.237**|**9.440**|8.866|**8.902**|
>     |FLUX.1 Kontext (dev)|0.256|0.225|7.236|**8.923**|6.998|
>     |Qwen-Image-Edit|0.263|0.232|9.111|8.789|8.567|
>     |**VAREdit-8B**|**0.266**|0.233|8.621|8.536|8.106|
>
> As a closed-source model, GPT-4o-Image performs best on both the two benchmarks due to the large scale of training data and model size.
> Among the open-source approaches, VAREdit-8B performs better than Step1X-Edit and FLUX.1 Kontext (dev) according to GPT-scores and a slightly lower than Qwen-Image-Edit (20B).
> Since our models adopt ~3.92M training data samples with 8.4B parameter size, we believe that an improved scaling and balance of the training dataset and model size will bring superior performance gain.

---

> ### Author Response · Authors · 2025-11-26
> **Author Rebuttal [2/2]**
>
> > **W4.**  This paper claims VAREdit as *the first tuning-based visual autoregressive model for instruction-guided image editing*, which is misleading. The authors might be expressing *image generation via next scale prediction*.
>
> Thanks for the suggestion. We have revised the expression by using "*among the first to introducenext-scale visual prediction into instruction-guided image editing*" in the revised manuscript. We have also added the description of NEP [1] in the related work section.
>
> > **Q1.** Provide the self-attention heatmaps under the SAR setting.
>
> Please refer to the response to **W1**.
>
> > **Q2.** Provide references to support the claim: Compared to diffusion models, AR approaches generally offer superior prompt adherence and faster inference.
>
> Thanks for the suggestion. We have revised the statement with some supporting references:
>
>     As AR approaches are well-suited for language modeling, they demonstrate a significant advantage in instruction-following performance [2][3] for image generation. Meanwhile, recent advances like hybrid parallelization and KV-scale mitigate the efficiency bottleneck [4][5]. These innovations are enhancing both generation quality and efficiency, enabling AR models to emerge as strong competitors to diffusion models, particularly in tasks requiring high fidelity to complex, compositional prompts.
>
> > **Q3.** Include performance comparisons with existing frontier editing models.
>
> Please refer to the response to **W3**.
>
> **References**
>
> [1] He Q, Chen X, Wang C, et al. Reasoning to Edit: Hypothetical Instruction-Based Image Editing with Visual Reasoning[J]. arXiv preprint arXiv:2507.01908, 2025.
>
> [2] Lai B, Juefei-Xu F, Liu M, et al. Unleashing in-context learning of autoregressive models for few-shot image manipulation[C]//Proceedings of the Computer Vision and Pattern Recognition Conference. 2025: 18346-18357.
>
> [3] Zhang Y, Li Y, Yang Y, et al. ReasonGen-R1: CoT for Autoregressive Image generation models through SFT and RL[J]. arXiv preprint arXiv:2505.24875, 2025.
>
> [4] Kumbong H, Liu X, Lin T Y, et al. HMAR: Efficient Hierarchical Masked Auto-Regressive Image Generation[C]//Proceedings of the Computer Vision and Pattern Recognition Conference. 2025: 2535-2544.
>
> [5] Li K, Chen Z, Yang C Y, et al. Memory-Efficient Visual Autoregressive Modeling with Scale-Aware KV Cache Compression[J]. arXiv preprint arXiv:2505.19602, 2025.

---

### Official Review · Reviewer_qmPC · 2025-10-30

**Soundness:** 3
**Presentation:** 3
**Contribution:** 2
**Rating:** 4
**Confidence:** 4

**Summary:**

This paper introduces VAREdit, an instruction-guided image editing framework that leverages a visual autoregressive modeling paradigm. By formulating editing as a next-scale prediction problem, VAREdit conditions on both source images and textual instructions to autoregressively generate multi-scale tokenized image features. The paper identifies a scale-mismatch issue in using finest-scale-only conditioning and proposes the Scale-Aligned Reference module, which bridges this gap by injecting scale-matched information only in the Transformer’s first self-attention layer. Experiments on EMU-Edit and PIE-Bench benchmarks demonstrate that VAREdit outperforms several competitive diffusion and AR-based baselines in both instruction adherence (as measured by GPT and CLIP scores) and inference speed. Additional ablations and qualitative studies illuminate the advantages of the SAR module.

**Strengths:**

1. The paper analyzes and highlights the limitations of both full-scale and finest-scale-only image conditioning for autoregressive editing tasks, a nuanced challenge not prominently addressed in prior work.
2. The proposed Scale-Aligned Reference module is conceptually well-motivated, solving the scale-mismatch issue in a resource-efficient way by using scale-aligned source features solely in the first attention layer. The supporting evidence includes explicit self-attention heatmaps and precise mathematical descriptions for how SAR modifies qkv calculation.
3. VAREdit sets a new state-of-the-art on the EMU-Edit and PIE-Bench datasets across a variety of metrics. Table 1 illustrates consistent improvements in both instruction adherence (GPT-Success, GPT-Balance) and region preservation (GPT-Over) compared to a broad selection of recent diffusion and AR-based methods.

**Weaknesses:**

1. The Related Work underappreciates or omits explicit discussion of closely related contemporaneous models for instruction-based or in-context image editing. [1,2,3] are especially relevant and directly relate to the manuscript’s focus and should be discussed in both the Related Work section and empirical comparisons if possible.

2. The approach is mostly evaluated in the standard fine-tuned regime and does not examine how VAREdit generalizes to unseen editing types, tasks, or user instructions without retraining—settings increasingly emphasized in recent autoregressive editing work[1]. The absence of in-context or few-shot adaptation benchmarks leaves a gap in understanding VAREdit’s flexibility beyond its tuned task distribution.

3. While qualitative figures suggest strong performance, a more honest (or even quantitative) exploration of limitations or failure modes (especially for global/text edits where 2.2B model underperforms) would strengthen the impact. For example, are there common instruction types, object classes, or image domains where even the SAR variant consistently underdelivers, and why? Are these failures more about model scale, data coverage, or something intrinsic to next-scale AR modeling?

4. All quantitative evaluation relies on CLIP and GPT-based metrics, which, while valuable, do not wholly replace exhaustive user studies, especially regarding over-editing artifacts or semantic misalignments. Some visual results imply subjective improvement, but explicit human preferences or error typologies are missing.

5. The experiments are mostly confined to two academic datasets. There is little examination of generalization to diverse, out-of-distribution, or real-world photographic content, which is critical for practical adoption. Even a small cross-dataset or wild-image ablation would be illuminating.

6. While the SAR module is intuitively described, the formulation around how scale-aligned features are constructed and integrated into the attention mechanism could be mathematically tighter. For example, clarification and explicit definition of how $\operatorname{Down}(\cdot)$ and its learned parameters are shared or adapted across scales would improve reproducibility. The transition from Equation 2 (cumulative features) to the SAR-enhanced self-attention mechanism could also use more technical rigor regarding masking, normalization, and gradient flow. Moreover, practical details such as how SAR interacts with positional encoding or codebook lookups for each scale should be more explicitly stated, especially since Figure 2 elides some of these corners.

7. While Figure 2 effectively visualizes the end-to-end pipeline, it omits some low-level components, leading to slight ambiguity (e.g., are the multi-scale VQ-Encoders shared across source and target? Are all tokens attended equally pre- and post-SAR? How are target and source regions aligned spatially or by edit mask, if at all?). Likewise, the heatmaps are highly informative, but a more interpretable color scale or clear highlighting of the causal directionality would help non-specialist readers.

8. The training dataset is aggregated and post-processed using a vision-language model filter, but only high-level processing details and a prompt are provided. Concrete numbers or public release status for the filtered data subsets would enhance reproducibility. Similarly, claims that code/models will be released “after acceptance” limit the immediate reproducibility of results.

9. Although the cited baselines are comprehensive for diffusion and AR models, the omission of multimodal LLM-based frameworks as competitors [2] weakens the claim of state-of-the-art. Even a partial or qualitative comparison would be helpful, as these systems are becoming increasingly prominent.

10. While the SAR mechanism is well-motivated, the extent to which it represents a substantial departure from architectural adjustments (as opposed to an incremental fix) is debatable. The key technical novelty (selective multi-scale reference injection in the first layer) may be seen as a relatively modest extension rather than a fundamentally new paradigm.

[1] Unleashing In-context Learning of Autoregressive Models for Few-shot Image Manipulation.

[2] Guiding Instruction-Based Image Editing via Multimodal Large Language Models

[3] Fireedit: Fine-grained instruction-based image editing via region-aware vision language model

**Questions:**

1. How does VAREdit perform in genuinely few-shot/in-context adaptation settings? Can the model apply novel editing instructions (or new types of edits) without SGD-based fine-tuning, or does it require full retraining? Comparative numbers or case studies against recent AR-based in-context editing models would be instructive.
2. What are the observed failure modes for VAREdit, especially for large/semantic/global edits or high-text-content regions? Can the authors provide a quantitative or qualitative assessment of these, and whether they are due to model scale, dataset limitations, or issues intrinsic to the SAR module?
3. Details on the SAR module’s construction: Are the downsampling mappings for scale-aligned features jointly learned, fixed, or derived from the tokenizer? How does SAR interact with residual connection and positional encoding in very deep Transformers?
4. How robust is VAREdit to highly ambiguous, underspecified, or conflicting instructions? Does it have mechanisms (e.g., uncertainty estimation, output confidence) to flag instructions where adherence is questionable?
5. Is there any plan for releasing the filtered and processed dataset used in training, and can more precise statistics be included in the public release for reproducibility?

---

> ### Author Response · Authors · 2025-11-26
> **Author Rebuttal [1/4]**
>
> Dear Reviewer qmPC,
>
>
> We appreciate the reviewer #qmPC acknowledging that our Scale-Aligned Reference (SAR) module is well-motivated and effectively solves the scale-mismatch issue. In response to your comments regarding robustness and clarity, we have expanded our analysis to include out-of-distribution generalization tests, explicit failure mode analysis, tighter mathematical formulations for the SAR module, etc.
>
> > **W1.**  The related work omits explicit discussion of closely related contemporaneous models of instruction-based or in-context image editing. Three relavent studies should be discussed or compared.
>
> Thanks for highlighting these contemporaneous works. We have updated **Section 2** (Related Work) to explicitly discuss these approaches and position VAREdit relative to them. Please refer to our responses to W9 for empirical comparisons of some advanced methods.
>
> > **W2.**  The study does not examing how VAREdit generalize to unseen editing types, task or user instructions without retraining.
> The absence of in-context or few-shot adaptation benchmarks leaves a gap in understanding VAREdit’s flexibility.
>
> We thank the reviewer for this insightful comment. While we acknowledge that in-context manipulation is a powerful emerging paradigm, VAREdit is explicitly designed for the zero-shot, instruction-driven setting. This paradigm provides a more intuitive user experience by relying solely on natural language commands and, through fine-tuning, achieves superior performance and instruction adherence compared to typical training-free approaches.
>
> To further evaluate VAREdit's generalization capability on unseen and sophisticated editing types, we have conducted additional qualitative experiments on highly imaginative scenarios that lie outside the standard training distribution. As detailed in the revised **Appendix A.5.2**, our model successfully handles complex tasks—such as applying Zaha Hadid’s architectural style or synthesizing semantic concepts (ice-flowing waterfalls, artistic sound waves, chess shadows, etc), demonstrating robust zero-shot adherence to novel instructions.
>
>
> > **W3.**  A more honest exploration of limitations or failure modes would strengthen the impact (e.g., failure patterns in common types, classes, image domains, along with the cause).
>
> Thanks for your suggestion. In the revised manuscript (**Appendix A.5.6**), we have provided a detailed analysis of common failure patterns. Our investigation reveals that while the 2.2B model struggles with specific categories like *Text Generation* and *Motion Change*, the 8.4B model largely mitigates these issues through scaling. However, even the larger model encounters difficulties in complex compositional tasks that require global modification and preservation of multiple distinct local elements. We attribute this limitation to the current density of processing scales $(h_k, w_k)$ and hypothesize that expanding both the image resolution and training data will resolve these fine-grained failures.
>
> > **W4.** All quantitative evaluation relies on CLIP and GPT scores, which do not wholy replace exhaustive user studies, especially regarding over-editing artifacts or semantic misalignments.
>
> Thanks for the suggestion. We fully agree that human evaluation is necessary for assessing subjective quality such as over-editing artifacts and semantic alignment. To mitigate this gap, we conducted a blind, randomized human preference study on the PIE-Bench dataset (700 samples).
>
> - **Study Design**: For each of the 700 samples in PIE-Bench, we presented participants with the source image, the text instruction, and the three edited images generated by our model (VAREdit-8B) and two strong baselines (Step1X-Edit, FLUX.1 Kontext (dev)). The three generated images were displayed in a randomized order to avoid positional bias.
>
> - **Task**: This is a blind, forced-choice study. Participants were asked to select the single best image that satisfies two criteria: (1) Instruction Following: How accurately the image reflects the textual instruction. (2) Image Quality: The realism of the edit and the absence of artifacts or over-editing.
>
> - **Participants**: We collected responses from 5 participants (e.g., graduate students in our field with experience in evaluating generative models).
> While we acknowledge this is a small-scale study due to the time constraints of the rebuttal period, the consistency in the results provides a strong and valuable signal regarding user preference.
>
> - **Results**: We report both the average number of wins and the overall preference rate for each model:
>
>     |Model|Avg. Win Count|Preference Rate (%)|GPT-Bal.|
>     |:--:|:--:|:--:|:--:|
>     |Step1X-Edit|210.4|30.1|7.351|
>     |FLUX.1 Kontext (dev)|237.4|33.9|6.998|
>     |**VAREdit-8B**|**252.2**|**36.0**|**8.106**|

---

> ### Author Response · Authors · 2025-11-26
> **Author Rebuttal [2/4]**
>
> (...**W4**) VAREdit-8B achieved the highest preference rate (36.0%), outperforming both strong competitors. This study confirms that VAREdit's quantitative strengths translate directly into superior perceptual quality for users. Further details are provided in our revised manuscript (**Appendix A.5.7**).
>
>
> > **W5.** The experiments are most confined to two academic datasets, without examination of generalization to diverse, OOD or real-world photographic content.
>
> Thank you for your suggestion. To rigorously test VAREdit's generalization to real-world and out-of-distribution (OOD) content, we have extended our evaluation to include ImgEdit (focusing on fine-grained, real-world instruction following) and GEdit-Bench (focusing on diverse and challenging editing types).
>
> - **Evaluation on ImgEdit-Bench** We focused on the Understanding-Grounding-Editing (UGE) suite, which contains harder, fine-grained samples compared to standard image editing tasks.
>
>     |Category|VAREdit-8B|VAREdit-2B|GPT-4o-Image|Step1X-Edit|ImgEdit-E1|UltraEdit|
>     |:--|:--:|:--:|:--:|:--:|:--:|:--:|
>     |Background Change|3.93|2.95|**4.62**|3.19|3.38|3.31|
>     |Object Extraction|1.52|1.94|**2.96**|1.87|2.55|2.02
>     |Attribute Alter|3.98|3.33|**4.26**|3.13|4.04|3.01|
>     |Style Transfer|4.60|4.05|**4.75**|4.44|4.38|3.69|
>     |Removement|**4.17**|3.51|3.81|2.61|2.40|1.71|
>     |Replacement|**4.54**|3.97|4.49|3.45|2.80|3.13|
>     |Addition|4.06|3.57|**4.65**|3.90|3.82|3.63|
>     |Motion Change|4.31|2.88|**4.76**|3.43|3.21|3.57|
>     |Hybrid Edit|3.30|2.57|**4.54**|2.52|2.87|2.33|
>     |UGE Score|4.09|2.81|**4.70**|3.11|3.20|2.36|
>
>     VAREdit-8B demonstrates robust generalization to real-world images, achieving the highest scores among open-source models in tasks like Removal, Replacement, and Motion Change. In the fine-grained UGE suit, it significantly outperforms baselines like Step1X-Edit (4.09 vs. 3.11). As a closed-source model, GPT-4o-Image achieves better performance with much larger training data and model size.
>
> - **Evaluation on GEdit-Bench** We first present the results across all categories of VAREdit-8B and VAREdit-2B (on English subset):
>
>     - Categorical Results on VAREdit-8B:
>
>         |Category| Intersection (G_SC) | Intersection (G_PQ) | Intersection (G_O) | Full (G_SC) | Full (G_PQ) | Full (G_O) |
>         |:--:|:--:|:--:|:--:|:--:|:--:|:--:|
>         |Background Change|7.862|5.793|6.433|7.875|5.950|6.480|
>         |Color Alter|9.000|6.265|7.320|9.050|6.600|7.557|
>         |Material Alter|7.500|5.714|6.215|7.300|5.700|6.016|
>         |Motion Change|5.864|6.591|5.249|4.725|6.550|4.067|
>         |PS Human|4.244|8.293|4.500|4.400|8.057|4.600|
>         |Style Change|7.771|5.958|6.572|7.483|5.983|6.450|
>         |Subject Add|8.237|6.842|7.400|7.917|6.933|7.132|
>         |Subject Remove|8.548|6.976|7.558|8.491|7.053|7.522|
>         |Subject Replace|8.261|6.348|6.989|8.050|6.217|6.777|
>         |Text Change|4.481|7.136|4.673|4.323|7.172|4.588|
>         |Tone Transfer|8.440|6.480|7.301|8.400|6.675|7.405|
>         |**Average**|7.292|6.581|6.383|7.092|6.626|6.236|
>
>     - Categorical Results on VAREdit-2B:
>
>         |Category| Intersection (G_SC) | Intersection (G_PQ) | Intersection (G_O) | Full (G_SC) | Full (G_PQ) | Full (G_O) |
>         |:--:|:--:|:--:|:--:|:--:|:--:|:--:|
>         |Background Change|5.793|4.897|4.749|5.925|4.700|4.832 |
>         |Color Alter|7.735|5.529|6.343|7.700|5.625|6.409 |
>         |Material Alter|4.714|4.071|3.982|4.625|4.500|3.952 |
>         |Motion Change|3.136|6.182|2.543|1.875|5.750|1.545 |
>         |PS Human|3.000|6.146|3.086|2.943|6.271|3.003 |
>         |Style Change|5.958|4.000|4.490|6.017|3.867|4.466 |
>         |Subject Add|6.500|6.079|5.800|6.267|5.783|5.616 |
>         |Subject Remove|6.857|6.333|6.290|6.632|6.281|6.132 |
>         |Subject Replace|6.500|4.696|5.135|6.450|4.767|5.191 |
>         |Text Change|1.778|5.383|2.293|1.980|5.525|2.412 |
>         |Tone Transfer|4.360|5.000|4.343|4.500|4.350|4.113 |
>         |**Average**|5.121|5.301|4.460|4.992|5.220|4.334 |
>
>     We then present the comparison results across representative approaches:
>
>     - Overall Comparison Results:
>
>         |Approach| Intersection (G_SC) | Intersection (G_PQ) | Intersection (G_O) | Full (G_SC) | Full (G_PQ) | Full (G_O) |
>         |:--:|:--:|:--:|:--:|:--:|:--:|:--:|
>         |InstructPix2Pix|3.335|6.210|3.234|3.296|6.189|3.219|
>         AnySD|3.122|5.865|2.919|3.053|5.882|2.854|
>         OmniGen|6.037|5.856|5.154|5.879|5.871|5.005|
>         Step1X-Edit|7.289|6.962|6.618|7.131|6.998|6.444|
>         GPT-4o-Image|**7.867**|**8.097**|**7.590**|**7.743**|**8.133**|**7.494**|
>         VAREdit-2B|5.121|5.301|4.460|4.992|5.220|4.334 |
>         VAREdit-8B|7.292|6.581|6.383|7.092|6.626|6.236 |
>
> Based on the fine-grained categorical results, VAREdit demonstrates exceptional advantages in Color Alteration, Subject Removal, Tone Transfer, and Subject Addition, consistently achieving high scores in both instruction adherence (G_SC) and perceptual quality (G_PQ).

---

> ### Author Response · Authors · 2025-11-26
> **Author Rebuttal [3/4]**
>
> (...**W5**) Regarding the relatively lower performance in Text Change and PS Human, we attribute this to specific gaps in data distribution. Specifically, our training set lacks the Chinese character samples present in the GEdit benchmark (e.g., transforming "大话西游" to "神话懂游") and contains limited coverage of face beautification data. We anticipate that scaling the training data to encompass these domains will effectively improve the performance.
>
> The aggregate results confirm that VAREdit-8B is highly competitive with Step1X-Edit and surpasses all other open-source baselines. It is worth noting that Step1X-Edit appears to benefit from a training distribution that is more closely aligned with the specific tasks found in GEdit-Bench.
>
> In addition, more qualitative case results have been presented in the revised manuscript (**Appendix A.5**), with more diverse editing types (e.g.,reasoning-based editing, multi-turn editing).
>
> > **W6.** The formulation around how scale-aligned feature are constructed and integrated into the attention mechanism could be mathematically tighter. Practical details should be more explicitly stated.
>
> Thanks. We have revised **Section 3 (Methodology)** and clarify the specific implementation details below:
>
> - Q6-1: How `Down()` and its learned parameters are shared or adapted across scales?
> - A6-1: `Down()` is a non-parameterized area pooling downsampling operation. It is implemented through `F.interpolate()` following previous studies (e.g., Infinity).
> - Q6-2: The transition from Equation 2 (cumulative features) to the SAR-enhanced self-attention mechanism could also use more technical rigor.
> - A6-2: The input cumulative features will be propagated through the positional encoding layer and the layer normalization before experience SAR in the first self-attention layers. These details are described in **Appendix A.4**.
> - Q6-3: How SAR interacts with positional encoding or codebook lookups for each scale?
> - A6-3: These details have been presented in **Appendix A.4**, where Equation 9 defines the vector quantization (BSQ), and Equation 10, 11 define the coordinates for source and target image scales within 2D-RoPE.
>
> For more detailed implementation, we invite the reviewer to check our released code (see **W8**).
>
> > **W7.** Figure 2 omits some low-level components, leading to slight ambiguity (e.g., are the multi-scale VQ-Encoders shared across source and target? Are all tokens attended equally pre- and post-SAR? How are target and source regions aligned spatially or by edit mask, if at all?). Likewise, the heatmaps could be improved with highlighting color scale or causal directionality.
>
> Many thanks to your valuable suggestions. To make it clear, we have revised the description in the new version of our manuscript (**Section 3, Methodology**). Specifically:
> - Q7-1: Are the multi-scale VQ-Encoders shared across source and target?
> - A7-1: Yes.
> - Q7-2: Are all tokens attended equally pre- and post-SAR?
> - A7-2: Yes. When calculating $\hat{\textbf{O}}_k^{(tgt)}$ at target scale $k$ (Equation 6), by replacing $\mathbf{K}_K^{(src)},\mathbf{V}_K^{(src)}$ with $\mathbf{K}_k^{(ref)},\mathbf{V}_k^{(ref)}$ in Equation 7, the output sequence length will not change.
> - Q7-3: How are target and source regions aligned spatially or by edit mask, if at all?
> - A7-3: To make the scale-aware alignment, we have employed 2D-RoPE to encode the spatial information of all input tokens before each self-attention module.
> The coordinate for the source and target token is presented in Equation 10 and 11 in **Appendix A.4**.
> In addition, a pretrained scale positional embedding is also integrated in the token representation.
> You can refer to our code repository to access further details (see our response to **W8**).
>
> Additionally, we have rebuilt the attention heatmaps by highlighting the left-bottom zone with different patterns between the first layer and other layers (**Figure 3**).
>
> > **W8.** Concrete numbers or public release status for the filtered data subsets would enhance reproducibility. The claim that code/models will be released after acceptance limit the immediate reproducibility.
>
> Thank you for your suggestion. After data pre-processing on SEED-Data-Edit, we obtain 2,919,318 training samples. We have organized the main training and inference code in an anonymized github repository: https://anonymous.4open.science/r/VAREdit-A5BB. Our results can be reproduced by initializing with the public Infinity text-to-image weights and training using the scripts provided in our repository.

---

> ### Author Response · Authors · 2025-11-26
> **Author Rebuttal [4/4]**
>
> > **W9.** The omission of multimodal LLM-based frameworks weakens the claim of state-of-the-art.
>
> Thanks. We have added more recent advanced approaches using multimodal LLM-based frameworks (e.g., Step1X-Edit, FLUX.1 Kontext (dev), Qwen-Image-Edit) for comprehensive evaluations.
>
> - Results on EMU-Edit:
>
>     |Approach|CLIP-Out.|CLIP-Dir.|GPT-Suc.|GPT-Over.|GPT-Bal.|
>     |:--:|:--:|:--:|:--:|:--:|:--:|
>     |Step1X-Edit|0.280|0.126|7.799|7.843|7.081|
>     |GPT-4o-Image|**0.286**|0.117|9.517|9.048|9.142|
>     |FLUX.1 Kontext (dev)|0.284|0.125|7.419|8.800|7.210|
>     |Qwen-Image-Edit|**0.286**|**0.133**|9.113|8.598|8.550|
>     |**VAREdit-8B**|0.282|0.125|8.284|8.461|7.892|
>
> - Results on PIE-Bench:
>
>     |Approach|CLIP-Whole|CLIP-Edit|GPT-Suc.|GPT-Over.|GPT-Bal.|
>     |:--:|:--:|:--:|:--:|:--:|:--:|
>     |Step1X-Edit|0.254|0.226|7.919|8.069|7.351|
>     |GPT-4o-Image|**0.266**|**0.237**|**9.440**|8.866|**8.902**|
>     |FLUX.1 Kontext (dev)|0.256|0.225|7.236|**8.923**|6.998|
>     |Qwen-Image-Edit|0.263|0.232|9.111|8.789|8.567|
>     |**VAREdit-8B**|**0.266**|0.233|8.621|8.536|8.106|
>
> We tried to also include MGIE as a baseline but the public released code did not work as expected. We leave this for future exploration.
>
> > **W10.** The extent of SAR to which it represents a substantial departure from architectural adjustments is debatable.
>
> Thanks. We agree that SAR alone is not a `fundamentally new paradigm` but an important architectural innovation that makes the visual autoregressive modeling paradigm computationally effective for image editing. Autoregressive models are mainly employed for image generation and there has been few works on applying large-scale autoregressive model to image editing. Directly applying autoregressive model to image editing is computation coslty and the SAR module is a key innovation to resolve this issue. We will clarify the scope of this contribution in the revised version.
>
> > **Q1.** How does VAREdit perform in genuinely few-shot/in-context adaptation settings? Can the model apply novel editing instructions (or new types of edits) without SGD-based fine-tuning, or does it require full retraining? Comparative numbers or case studies against recent AR-based in-context editing models would be instructive.
>
> Please refer to the response to **W2**.
>
> > **Q2.** What are the observed failure modes for VAREdit, especially for large/semantic/global edits or high-text-content regions?
>
> Please refer to the response to **W3**.
>
> > **Q3.** Details on the SAR module’s construction: Are the downsampling mappings for scale-aligned features jointly learned, fixed, or derived from the tokenizer? How does SAR interact with residual connection and positional encoding in very deep Transformers?
>
> The `Down()` is a non-parameterized area pooling downsampling operation. It is implemented through `F.interpolate()` following previous studies (e.g., Infinity).
> Since SAR is only adopted in the first self-attention layers, the propagation in deeper Transformer layers remain the original next-scale prediction paradigm.
>
> > **Q4.** How robust is VAREdit to highly ambiguous, underspecified, or conflicting instructions? Does it have mechanisms (e.g., uncertainty estimation, output confidence) to flag instructions where adherence is questionable?
>
> Thanks for your question.
> To address this issue, we have conducted a qualitative study on several representative samples with ambiguous or conflicted instructions. The results can be accessed in the revised manuscript (**Appendix A.5.5**). To summarize, VAREdit can handle ambiguous instructions to some extent but it may not be robust to highly ambiguous, underspecified, or conflicting instructions.
>
>
> > **Q5.** Is there any plan for releasing the filtered and processed dataset used in training, and can more precise statistics be included in the public release for reproducibility?
>
> Please refer to the response to **W8**.
>
>
> **References**
>
> [1] Fu T J, Hu W, Du X, et al. Guiding instruction-based image editing via multimodal large language models[C]. In ICLR, 2024.

---

> > ### Comment · Reviewer_qmPC · 2025-11-27
> >
> > The authors have addressed most of my concerns. I will raise my score and support the acceptance of this paper.

---

> ### Author Response · Authors · 2025-11-28
> **Thanks for your positive rating!**
>
> Dear Reviewer qmPC,
>
> Thank you very much for your positive rating! We sincerely appreciate your time and effort in sharing such valuable feedback with us, and your positive attitude towards our work.
> We are glad that our rebuttal has addressed most of your concerns, and we are happy to discuss if you have further questions.
>
> Best regards!
>
> Authors of Paper 15382

---

### Official Review · Reviewer_Hejp · 2025-10-31

**Soundness:** 3
**Presentation:** 3
**Contribution:** 3
**Rating:** 6
**Confidence:** 3

**Summary:**

This paper introduces VAREdit, the first training-based Visual Autoregressive (VAR) model for instruction-guided image editing. The key contribution is reframing image editing as a next-scale prediction problem, where the model autoregressively generates multi-scale target features conditioned on source image features and textual instructions. The authors identify a critical scale-mismatch challenge when conditioning on finest-scale source features alone: high-frequency details disrupt the prediction of coarse-grained target structures. To address this, they propose the Scale-Aligned Reference (SAR) module, which dynamically generates and injects scale-matched source information specifically into the first self-attention layer, while deeper layers continue to use finest-scale conditioning. Through systematic analysis of self-attention patterns, the authors demonstrate that this design aligns with the model's functional differentiation across layers—where the first layer establishes global structure and deeper layers perform local refinement. Extensive experiments on EMU-Edit and PIE-Bench benchmarks show that VAREdit achieves substantial improvements over state-of-the-art diffusion-based methods

**Strengths:**

1. This paper explores how to apply VAR architecture to image editing, moving away from dominant diffusion approaches. The SOTA results on balanced metrics and 2.2x speedup are significant, suggesting AR models are a highly promising direction for image editing.
2. The core contribution is the deep analysis of the scale mismatch problem in VAR conditioning. The attention map analysis (Figure 3) is strong evidence. The resulting SAR module is an elegant and efficient solution that precisely targets the identified bottleneck (the first layer) without incurring extra inference cost.
3. The paper rightly identifies flaws in standard metrics. Relying on GPT-Balance and especially the analysis in Figure 4 (evaluating preservation only on successful edits) provides a much more rigorous and convincing evaluation than prior work.

**Weaknesses:**

1. The paper's primary metrics (GPT-Suc., GPT-Over., GPT-Bal.) rely on GPT-4o as a judge. While arguably better than CLIP, this is costly, slow, and dependent on a proprietary API, making the evaluation results difficult and expensive to reproduce. The authors should provide an alternative using an open-source VLM (such as Qwen3-VL) in the rebuttal.
2. The authors used CLIP and GPT scores as evaluation metrics, but lacked test results on benchmarks commonly used in the editing field, such as ImgEdit and GEdit-Bench. Considering the authors' statement that CLIP scores do not accurately reflect editing performance, and the overhead and uncertainty of GPT scores, using ImgEdit and GEdit-Bench can provide reproducible and stable comparative results.
3. The training data was filtered using an external VLM (Kimi-VL), which discarded ~1M samples. This is a significant, non-reproducible step, and the final model quality may be heavily dependent on this external filter.

**Questions:**

None

---

> ### Author Response · Authors · 2025-11-26
> **Author Rebuttal [1/2]**
>
> Dear Reviewer Hejp,
>
> We sincerely thank #Hejp for recognizing the significant speedup and the novelty of our SAR module. To address your concerns regarding reproducibility and benchmarking, we have integrated the open-source Qwen3-VL metric and conducted further evaluations on the ImgEdit and GEdit benchmarks.
>
> > **W1.**  The paper's primary metrics rely on GPT-4o, which is costly and slow. The authors should provide an alternative using an open-source VLM (such as Qwen3-VL).
>
> We appreciate the suggestion to enhance evaluation reproducibility and transparency. Following your recommendation, we adopted the open-source `Qwen3-VL-30B-A3B-Instruct` as an alternative evaluator. Using the identical prompts as our GPT-4o evaluation, we assessed the Success (Qwen-Suc.), Over-edit (Qwen-Over.), and Balance (Qwen-Bal.) scores for all approaches.
>
> We present the side-by-side comparison of GPT-based and Qwen-based metrics below:
>
> - Results on EMU-Edit:
>
>     |Approach|GPT-Suc.|GPT-Over.|GPT-Bal.|Qwen-Suc.|Qwen-Over.|Qwen-Bal.|
>     |:---:|:---:|:---:|:---:|:---:|:---:|:---:|
>     |InstructPix2Pix|3.358|6.299|2.923|4.660|3.754|2.722|
>     |UltraEdit|4.881|7.704|4.541|6.089|4.253|4.083|
>     |OmniGen|4.738|**8.709**|4.666|5.714|3.160|3.233|
>     |AnySD|3.098|8.590|3.129|4.307|2.816|2.324|
>     |EditAR|3.582|7.260|3.305|4.364|4.120|2.980|
>     |ACE++|2.375|5.979|2.076|3.639|3.843|2.147|
>     |ICEdit|5.027|7.591|4.785|6.309|4.146|3.672|
>     |**VAREdit-2B**|6.210|7.055|5.662|7.272|4.330|4.068|
>     |**VAREdit-8B**|**8.284**|8.461|**7.892**|**8.858**|**4.612**|**5.273**|
>
> - Results on PIE-Bench:
>
>     |Approach|GPT-Suc.|GPT-Over.|GPT-Bal.|Qwen-Suc.|Qwen-Over.|Qwen-Bal.|
>     |:---:|:---:|:---:|:---:|:---:|:---:|:---:|
>     |InstructPix2Pix|4.794|6.534|4.034|5.483|4.419|3.503|
>     |UltraEdit|5.831|8.350|5.580|6.071|4.603|4.400|
>     |OmniGen|3.459|**8.939**|3.498|3.921|2.681|2.511|
>     |AnySD|3.456|7.806|3.326|4.067|3.686|2.558|
>     |EditAR|5.070|8.116|4.707|5.521|4.467|4.009|
>     |ACE++|2.743|8.093|2.574|3.294|3.663|2.447|
>     |ICEdit|5.321|7.593|4.933|6.070|4.539|3.888|
>     |**VAREdit-2B**|7.530|8.083|6.996|7.960|5.029|5.252|
>     |**VAREdit-8B**|**8.621**|8.536|**8.105**|**9.051**|**5.461**|**6.165**|
>
> The results confirm that VAREdit's superiority is robust to the choice of VLM evaluator. While absolute scores shift due to differences in VLM calibration, the relative rankings remain consistent: VAREdit-8B maintains the top position in the Balance metric on both benchmarks.
>
> > **W2.**  This paper lacks evaluation on commonly used benchmarks like ImgEdit and GEdit-Bench as reproducible and stable comparative results.
>
>
> We appreciate the suggestion to broaden our evaluation scope beyond existing EMU-Edit and PIE-Bench. In direct response to your suggestion, we have extended our experimental evaluation to include ImgEdit and GEdit-Bench, comparing VAREdit against representative approaches.
>
>
> - **Results on ImgEdit-Bench** We evaluated performance across all categories in the Basic-Edit suite and the Understanding-Grounding-Editing (UGE) suite. The results are summarized below:
>
>     |Category|VAREdit-8B|VAREdit-2B|GPT-4o-Image|Step1X-Edit|ImgEdit-E1|UltraEdit|
>     |:--|:--:|:--:|:--:|:--:|:--:|:--:|
>     |Background Change|3.93|2.95|**4.62**|3.19|3.38|3.31|
>     |Object Extraction|1.52|1.94|**2.96**|1.87|2.55|2.02
>     |Attribute Alter|3.98|3.33|**4.26**|3.13|4.04|3.01|
>     |Style Transfer|4.60|4.05|**4.75**|4.44|4.38|3.69|
>     |Removement|**4.17**|3.51|3.81|2.61|2.40|1.71|
>     |Replacement|**4.54**|3.97|4.49|3.45|2.80|3.13|
>     |Addition|4.06|3.57|**4.65**|3.90|3.82|3.63|
>     |Motion Change|4.31|2.88|**4.76**|3.43|3.21|3.57|
>     |Hybrid Edit|3.30|2.57|**4.54**|2.52|2.87|2.33|
>     |UGE Score|4.09|2.81|**4.70**|3.11|3.20|2.36|
>
> While the closed-source GPT-4o-Image achieves high scores across most categories, VAREdit-8B emerges as the top-performing open-source model. It achieves the best results among open-source contenders in key categories such as Background Change, Style Transfer, Removal, Replacement, and Motion Change. Notably, in fine-grained tasks (UGE Score), VAREdit-8B significantly outperforms competitive baselines like Step1X-Edit and UltraEdit (4.09 vs. 3.11/2.36).

---

> ### Author Response · Authors · 2025-11-26
> **Author Rebuttal [2/2]**
>
> (...**W2**)
> - **Results on GEdit-Bench** We further evaluated VAREdit-8B and VAREdit-2B on English subset of GEdit-Bench across diverse editing categories.
>
>     - Categorical Results (VAREdit-8B):
>
>         |Category| Intersection (G_SC) | Intersection (G_PQ) | Intersection (G_O) | Full (G_SC) | Full (G_PQ) | Full (G_O) |
>         |:--:|:--:|:--:|:--:|:--:|:--:|:--:|
>         |Background Change|7.862|5.793|6.433|7.875|5.950|6.480|
>         |Color Alter|9.000|6.265|7.320|9.050|6.600|7.557|
>         |Material Alter|7.500|5.714|6.215|7.300|5.700|6.016|
>         |Motion Change|5.864|6.591|5.249|4.725|6.550|4.067|
>         |PS Human|4.244|8.293|4.500|4.400|8.057|4.600|
>         |Style Change|7.771|5.958|6.572|7.483|5.983|6.450|
>         |Subject Add|8.237|6.842|7.400|7.917|6.933|7.132|
>         |Subject Remove|8.548|6.976|7.558|8.491|7.053|7.522|
>         |Subject Replace|8.261|6.348|6.989|8.050|6.217|6.777|
>         |Text Change|4.481|7.136|4.673|4.323|7.172|4.588|
>         |Tone Transfer|8.440|6.480|7.301|8.400|6.675|7.405|
>         |**Average**|7.292|6.581|6.383|7.092|6.626|6.236|
>
>     - Categorical Results (VAREdit-2B):
>
>         |Category| Intersection (G_SC) | Intersection (G_PQ) | Intersection (G_O) | Full (G_SC) | Full (G_PQ) | Full (G_O) |
>         |:--:|:--:|:--:|:--:|:--:|:--:|:--:|
>         |Background Change|5.793|4.897|4.749|5.925|4.700|4.832 |
>         |Color Alter|7.735|5.529|6.343|7.700|5.625|6.409 |
>         |Material Alter|4.714|4.071|3.982|4.625|4.500|3.952 |
>         |Motion Change|3.136|6.182|2.543|1.875|5.750|1.545 |
>         |PS Human|3.000|6.146|3.086|2.943|6.271|3.003 |
>         |Style Change|5.958|4.000|4.490|6.017|3.867|4.466 |
>         |Subject Add|6.500|6.079|5.800|6.267|5.783|5.616 |
>         |Subject Remove|6.857|6.333|6.290|6.632|6.281|6.132 |
>         |Subject Replace|6.500|4.696|5.135|6.450|4.767|5.191 |
>         |Text Change|1.778|5.383|2.293|1.980|5.525|2.412 |
>         |Tone Transfer|4.360|5.000|4.343|4.500|4.350|4.113 |
>         |**Average**|5.121|5.301|4.460|4.992|5.220|4.334 |
>
>     - Overall Comparison:
>
>         |Approach| Intersection (G_SC) | Intersection (G_PQ) | Intersection (G_O) | Full (G_SC) | Full (G_PQ) | Full (G_O) |
>         |:--:|:--:|:--:|:--:|:--:|:--:|:--:|
>         |InstructPix2Pix|3.335|6.210|3.234|3.296|6.189|3.219|
>         AnySD|3.122|5.865|2.919|3.053|5.882|2.854|
>         OmniGen|6.037|5.856|5.154|5.879|5.871|5.005|
>         Step1X-Edit|7.289|6.962|6.618|7.131|6.998|6.444|
>         GPT-4o-Image|**7.867**|**8.097**|**7.590**|**7.743**|**8.133**|**7.494**|
>         VAREdit-2B|5.121|5.301|4.460|4.992|5.220|4.334 |
>         VAREdit-8B|7.292|6.581|6.383|7.092|6.626|6.236 |
>
>
> Based on the fine-grained categorical results, VAREdit demonstrates exceptional advantages in Color Alteration, Subject Removal, Tone Transfer, and Subject Addition, consistently achieving high scores in both instruction adherence (G_SC) and perceptual quality (G_PQ).
>
> Regarding the relatively lower performance in Text Change and PS Human, we attribute this to specific gaps in data distribution. Specifically, our training set lacks the Chinese character samples present in the GEdit benchmark (e.g., transforming "大话西游" to "神话懂游") and contains limited coverage of face beautification data. We anticipate that scaling the training data to encompass these domains will effectively improve the performance.
>
> The aggregate results confirm that VAREdit-8B is highly competitive with Step1X-Edit and surpasses all other open-source baselines. It is worth noting that Step1X-Edit appears to benefit from a training distribution that is more closely aligned with the specific tasks found in GEdit-Bench.
>
> > **W3.**  The training data filtering process using Kimi-VL is significant and non-reproducible, making the model quality heavily depends on this filter.
>
> We sincerely appreciate the reviewer's attention to reproducibility. We would like to clarify that the filtering pipeline is reproducible and accessible. First, we utilized the open-source Kimi-VL model (Kimi-VL-A3B-Thinking), ensuring the filtering logic is available to the community. Second, the computational cost for this process is manageable, requiring 8 x A800 GPUs for less than two days. To further facilitate exact reproducibility and save computational resources for future researchers, we will publicly release the full JSON file containing the indices of the 2,919,318 "cleaned" samples used in our training.

---

### Author Response · Authors · 2025-12-04
**Author Summary of Rebuttal and Paper Revision**

Dear PCs, SACs, ACs and Reviewers,

We sincerely thank you for your time, effort and constructive feedback on our work.
In response to your insightful suggestions, we have performed substantial new experiments, conducted in-depth analyses, and carefully revised our manuscript (revisions are marked in blue for clarity).

To facilitate the final review process and assist the ACs, we summarize the reviewers' key concerns and our corresponding updates below.

1. Substantial Experimental Updates

    1a. **Evaluation with Open-Source VLMs** [Reviewer Hejp]: To address the reliance on proprietary models, we have incorporated the open-source Qwen3-VL as an alternative evaluator, diversifying our assessment.

    1b. **Expanded Benchmark Evaluation** [Reviewer Hejp, dF8r]: We have extended our evaluation on two additional benchmarks, ImgEdit and GEdit-Bench, to demonstrate the broader applicability and robustness of our method.

    1c. **Comparison with Advanced Methods** [Reviewer jZVb]: We included a comparative analysis against four leading contemporary models: Step1X-Edit, GPT-4o-Image, FLUX.1 Kontext (dev) and Qwen-Image-Edit, further contextualizing our performance.

    1d. **Additional Qualitative Results** [Reviewer qmPC, dF8r]: The revised manuscript includes a new set of qualitative results showcasing imaginative editing, reasoning-based editing, multi-turn interactions, and a transparent analysis of failure cases.

    1e. **Controlled User Study** [Reviewer qmPC, jZVb]: To provide more intuitive and reliable comparisons, a controlled human evaluation has been conducted on PIE-Bench.

2. Implementation and Model Analysis

    2a. **Reproducibility** [Reviewer Hejp, qmPC]: We have clarified the data filtering process and released the core code to ensure reproducibility.

    2b. **Attention Visualization of SAR** [Reviewer jZVb]: We provided the corresponding self-attention heatmaps of the SAR variant to demonstrate the scale-matched reference.

3. Context Revisions and Clarifications

    3a. **Expanded Related Work** [Reviewer qmPC, jZVb]: The related work section has been updated to include and discuss the suggested literature.

    3b. **Technical Clarifications** [Reviewer qmPC]: We have addressed specific queries by clarifying technical details.

    3c. **Refined Phrasing and Claims** [Reviewer jZVb]: We have carefully revised certain statements to prevent potential misinterpretation and ensure precision.

Once again, we sincerely thank all the reviewers, ACs, SACs and PCs for their invaluable feedback, which has significantly strengthened the quality of our paper.

Best regards,

Authors of Paper 15382

---

### Meta-Review · Area_Chair_FF8T · 2025-12-31

**Summary:**

This paper proposes VAREdit, a visual autoregressive modeling framework for instruction-guided image editing, aiming to enhance performance in complex editing tasks. Reviewers acknowledge the work's innovation in visual autoregressive modeling and its potential for instruction-guided image editing, while raising concerns about evaluation reliability, reproducibility, experimental coverage, and technical clarity.

Most of critical concerns are resolved via substantial experimental updates, supplementary analyses, and added explanations. Concern about over-reliance on costly and slow GPT-4o metrics, and lack of evaluations on common benchmarks (ImgEdit, GEdit-Bench) are addressed. Inquiries regarding omitted discussions of contemporary models, insufficient generalization to real-world content, lack of limitation analysis, and inadequate comparisons with frontier models are effectively resolved. Concern about missing self-attention heatmap analysis and lack of controlled human studies are fully addressed.

However some concerns are not sufficiently resolved despite supplementary explanations. For the opacity of the training data filtering process using Kimi-1s, while the authors mentioned the process, they did not provide detailed, reproducible steps or parameters, leaving potential barriers to model replication. Besides, for the structural details of the SAR model, the authors directed reviewers to refer to the upcoming code but did not clearly elaborate on how the SAR model interacts with residual connections and positional encoding in deep Transformers in the manuscript, lacking sufficient technical clarification. The authors need to provide further revisions in the final version.

Based on the above considerations, I think the current manuscript basically meets the requirements of ICLR, and I suggest considering accepting this paper.

**Reviewer Concerns:**

Most of critical concerns are resolved via substantial experimental updates, supplementary analyses, and added explanations. The concern about opacity of the training data filtering process using Kimi-1s is not fully addressed. While the authors mentioned the process, they did not provide detailed, reproducible steps or parameters, leaving potential barriers to model replication. The concern about structural details of the SAR model is not fully addressed. The authors directed reviewers to refer to the upcoming code but did not clearly elaborate on how the SAR model interacts with residual connections and positional encoding in deep Transformers in the manuscript, lacking sufficient technical clarification.

**Reviewer Scores:**

Reviewer qmPC explicitly stated that most concerns have been addressed and expressed an intention to raise the score. Thus, the score is likely to be increased. Reviewer jZVb may raise the score and other reviewers may keep their positive scores.

---

### Decision · Program_Chairs · 2026-01-26

Accept (Poster)